# Energy-based generator matching:
# A neural sampler for general state space

**Dongyeop Woo**[1][*]    **Minsu Kim**[1,2]    **Minkyu Kim**[1]    **Kiyoung Seong**[1]    **Sungsoo Ahn**[1]

[1]Korea Advanced Institute of Science and Technology (KAIST)    [2]Mila - Quebec AI Institute

## Abstract

We propose *Energy-based generator matching (EGM)*, a modality-agnostic approach to train generative models from energy functions in the absence of data. Extending the recently proposed generator matching, EGM enables training of arbitrary continuous-time Markov processes, e.g., diffusion, flow, and jump, and can generate data from continuous, discrete, and a mixture of two modalities. To this end, we propose estimating the generator matching loss using self-normalized importance sampling with an additional bootstrapping trick to reduce variance in the importance weight. We validate EGM on both discrete and multimodal tasks up to 100 and 20 dimensions, respectively.

## 1   Introduction

We tackle the problem of drawing samples from a Boltzmann distribution $p_{\text{target}}(x) \propto \exp(-\mathcal{E}(x))$ given only oracle access to the energy function $\mathcal{E}(x)$ and no pre-computed equilibrium samples. Such "energy-only" inference arises throughout machine learning, e.g., Bayesian inference [15] and statistical physics, e.g., computing thermodynamic averages [29], yet it remains intractable in high-dimensional or combinatorial state spaces. Classical methods based on Markov processes, e.g., Metropolis–Hastings [28, 18], Gibbs sampling [16], and Metropolis-adjusted Langevin algorithm [17, 33], provide asymptotically exact guarantees, but at the cost of long mixing times. In practice, one must run chains for a vast number of steps to traverse metastable regions, and each transition incurs an expensive energy evaluation, limiting applicability to large-scale systems [32].

Deep generative models, particularly diffusion and flow models tied to continuous-time processes [36, 25], offer a compelling alternative: after training, a cheap constant-cost network evaluation can produce independent samples. Once trained, they do not require energy queries for inference. These models show great success in vision [34], language [24], and audio [21], and can be transferred to unseen conditions. However, their success critically depends on data-driven training on a large number of true equilibrium samples, which are unavailable when only an energy function is known.

In response, researchers have designed new energy-driven algorithms, coined diffusion samplers, to train diffusion-based generative models to sample from the Boltzmann distribution. For example, path integral samplers (PIS) [42] and denoising diffusion samplers (DDS) [38] minimize the divergence between a forward SDE and the backward SDE defined by a Brownian bridge. Akhound-Sadegh et al. [1] proposed iterated denoising energy matching (iDEM), a simulation-free approach to train diffusion models, highlighting the expensive simulation procedure for acquiring new samples from the diffusion-based models. Generative flow networks (GFNs) [3, 22], originating from reinforcement learning, can also be interpreted as a continuous-time Markov process (CTMP) in the limit [6].

However, there still exists a gap between the data- and energy-driven training schemes, particularly on the choice of state spaces and CTMP. Especially, Holderrieth et al. [19] recently proposed generator

---

[*]Correspondence to: dongyeop.woo@kaist.ac.kr

39th Conference on Neural Information Processing Systems (NeurIPS 2025).

matching (GM), which allows unified data-driven training of continuous-time processes ranging from flow, diffusion, and jump processes for both continuous and discrete state spaces. This allows training a generative model for hybrid data, e.g. amino acid sequences and 3D coordinates of a protein [7], and greatly expands the available space of parameterization.

**Contribution.** We propose energy-based generator matching (EGM), an energy-driven training framework for continuous-time Markov processes parameterized by a neural network. EGM accommodates continuous, discrete, and mixed state spaces and applies to various process types, including diffusion-, flow-, and jump-based models. Our work greatly expands the scope of energy-driven training for continuous-time neural processes (See Table 1).

Table 1: Comparison of sampling methods by state space type and underlying Markov process. "D", "F", "J", "J (D)" denotes the diffusion, flow, continuous jump, and discrete jump process respectively.

| Method | D | F | J | J (D) |
|---|---|---|---|---|
| PIS [42], iDEM [1], CMCD [39] | ✔ | ✘ | ✘ | ✘ |
| iEFM [40], LFIS [37], NFS² [8] | ✘ | ✔ | ✘ | ✘ |
| LEAPS [20] | ✘ | ✘ | ✘ | ✔ |
| **EGM (ours)** | ✔ | ✔ | ✔ | ✔ |

To this end, our EGM estimates the generator matching loss with self-normalized importance sampling, which requires approximating samples from the target distribution. However, this is challenging due to the high variance in importance weights. To alleviate this issue, we introduce bootstrapping, which allows the generator to be estimated using easy-to-approximate samples from the nearby future time step. It reduces the variance of importance weights, significantly boosting the sampler's performance.

We validate our work through experiments on various target distributions: discrete Ising model and three joint discrete-continuous tasks, i.e., the Gaussian-Bernoulli restricted Boltzmann machine (GB-RBM) [9], joint double-well potential (JointDW4), and joint mixture of Gaussians (JointMoG). Our findings demonstrate that the training algorithm enables parameterized CTMP to learn the desired distribution and is scalable to reasonably-sized problems.

## 2 Preliminary: Generator matching on general state space

In this section, we provide preliminaries on generator matching (GM) [19], which allows generative modeling using arbitrary continuous-time Markov processes (CTMP).

**Notation.** Let $S$ denote the state space with a reference measure $\nu$. We let $p_{\text{target}}$ denote the target distribution with samples $x_1 \sim p_{\text{target}}$. We denote a probability measure $p$ on $S$ by $p(dx)$, where "$dx$" indicates integration with respect to $p$ in the variable $x$. If $p$ admits a density, and when no confusion arises, we use $p$ both for the measure $p(dx)$ and its density $p(x) := \frac{dp}{d\nu}(x)$ with respect to $\nu$.

**Overview of generator matching.** GM relies on a set of time-varying probability distributions $(p_{t|1}(dx|x_1))_{0 \le t \le 1}$ depending on a data point $x_1 \in S$, coined a conditional probability path. This induces a corresponding marginal probability path $(p_t)_{0 \le t \le 1}$ via the hierarchical sampling procedure:

$$x_1 \sim p_{\text{target}}, \ x \sim p_{t|1}(dx|x_1) \quad \Rightarrow \quad x \sim p_t(dx). \tag{1}$$

GM trains a Markov process $(X_t^\theta)_{0 \le t \le 1}$, parameterized by a neural network, to match the marginal probability path $p_t$. After training, the induced probability path of the Markov process aligns with $p_t$, enabling sampling from $p_{\text{target}} = p_1$ by simulating the trained process $X_t^\theta$.

**Generators.** A generator characterizes a CTMP through the expected change of a test function $f : S \to \mathbb{R}$ in an infinitesimal time frame. It is a linear operator defined as:

$$\mathcal{L}_t f(x) = \frac{d}{dh}\bigg|_{h=0} \mathbb{E}_{X_{t+h} \sim p_{t+h|t}(\cdot|x)}[f(X_{t+h})] = \lim_{h \to 0} \frac{\mathbb{E}_{X_{t+h} \sim p_{t+h|t}(\cdot|x)}[f(X_{t+h})] - f(x)}{h}, \tag{2}$$

where $p_{t+h|t}$ denotes the transition kernel of Markov process $X_t$. If two Markov processes $X^1$ and $X^2$ have identical generators $\mathcal{L}_t^1$ and $\mathcal{L}_t^2$, the processes are equivalent.

**Linear parametrization of generators.** For commonly used Markov processes, e.g., diffusion, flow, and jump processes, the generator can be linearly parameterized as $\mathcal{L}_t f(x) = \langle \mathcal{K}f(x), F_t(x) \rangle_V$ where $\mathcal{K}$ is an operator fixed for each type of Markov process, $V$ is a vector space, and $F_t(x) \in V$ is the parameterization. For example, $F_t = u_t$ for flow model with ODE $dx = u_t(x)dt$, $F_t = \sigma_t^2$ for diffusion with SDE $dx = \sigma_t(x)dw$, or $F_t = Q_t$ for discrete jump with transition rate matrix $Q_t(y, x)$ described by the Kolmogorov equation $\partial_t p_t(y) = \sum_{y \in S} Q_t(y, x) p_t(x)$.

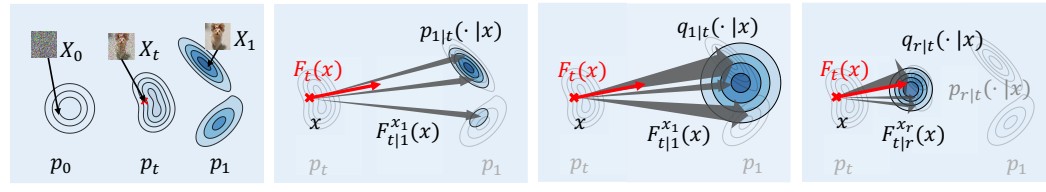

| (a) Probability path | (b) Generator matching | (c) EGM | (d) EGM w/ bootstrapping |

Figure 1: Overview of energy-based generator matching (EGM). (a) The target probability path that interpolates between the prior and the target distribution; we aim to estimate the $F_t(x)$ as a (weighted) average of conditional generators. (b) GM draws $x_1 \sim p_{1|t}(\cdot|x)$ with uniformly weighted $F_{t|1}^{x_1}(x)$. (c) EGM draws $x_1 \sim q_{1|t}(\cdot|x)$ with importance weighted $F_{t|1}^{x_1}(x)$. (d) EGM w/ bootstrapping draws $x_r \sim q_{r|t}(\cdot|x)$ with importance weighted $F_{t|r}^{x_r}(x)$.

**Marginalization trick.** The key idea of GM is to express a marginal generator $\mathcal{L}_t$ that generates probability path $p_t$ by conditional generator $\mathcal{L}_{t|1}^{x_1}$ for the conditional probability path $p_{t|1}(\cdot|x_1)$. This leads to the expression of the parameterization $F_t$ for the generator $\mathcal{L}_t$ using parameterization $F_{t|1}^{x_1}$ of conditional generator $\mathcal{L}_{t|1}^{x_1}$. To be specific, the marginalization trick is expressed as:

$$\mathcal{L}_t f(x) = \mathbb{E}_{x_1 \sim p_{1|t}(\cdot|x)}\big[\mathcal{L}_{t|1}^{x_1} f(x)\big], \qquad F_t(x) = \mathbb{E}_{x_1 \sim p_{1|t}(\cdot|x)}\big[F_{t|1}^{x_1}(x)\big], \qquad (3)$$

where $p_{1|t}(dx_1|x)$ is the posterior distribution, i.e., the conditional distribution over data $x_1$ given an observation $x$ at time $t$. Intuitively, at any point $(x, t)$, the marginal generator $\mathcal{L}_t$ steers $x$ in the average direction of endpoints $x_1$ sampled from the target distribution.

**Conditional generator matching.** GM trains a neural network $F_t^\theta$ to approximate the parametrization $F_t$ of marginal generator $\mathcal{L}_t$ using a Bregman divergence $D : V \times V \to \mathbb{R}_{\geq 0}$:

$$L_{\text{GM}}(\theta) = \mathbb{E}_{t \sim \text{Unif}, \, x_t \sim p_t}\big[D\big(F_t(x_t), F_t^\theta(x_t)\big)\big], \qquad (4)$$

which is hard to minimize since $F_t$ is intractable to estimate. Instead, one can minimize the conditional generator matching (CGM) loss expressed as follows:

$$L_{\text{CGM}}(\theta) = \mathbb{E}_{t \sim \text{Unif}, \, x_1 \sim p_{\text{target}}, \, x_t \sim p_{t|1}(\cdot|x_1)}\big[D\big(F_{t|1}^{x_1}(x_t), F_t^\theta(x_t)\big)\big]. \qquad (5)$$

One can derive that $\nabla_\theta L_{\text{GM}}(\theta) = \nabla_\theta L_{\text{CGM}}(\theta)$, and gradient-based minimization of the CGM loss is equivalent to that of the GM loss.

## 3 Energy-based generator matching

In this section, we introduce energy-based generator matching (EGM), the first method for learning neural samplers on general state spaces via continuous-time Markov processes. EGM extends the GM framework to energy-driven training by estimating the marginal generator with self-normalized importance sampling (SNIS). To this end, we define a variant of GM loss $L_{\text{GM}}$ in Equation (4):

$$L_{\text{EGM}}(\theta) = \mathbb{E}_{t \sim \text{Unif}, \, x_t \sim p_t^{\text{ref}}}\big[D\big(\hat{F}_t(x_t), F_t^\theta(x_t)\big)\big], \qquad (6)$$

where $\hat{F}_t$ is the energy-driven estimator of the parametrization $F_t$ and $p_t^{\text{ref}}$ is a reference distribution whose support covers the probability path $p_t$. Note that, when $\hat{F}_t = F_t$, minimizing $L_{\text{EGM}}$ guarantees that the learned sampler $F_t^\theta$ recovers samples from the target distribution $p_{\text{target}}$.

We first present our scheme to compute the estimator $\hat{F}_t$, then describe a bootstrapping technique for variance reduction of the importance weight, and finally provide the training algorithm. Figure 1 provides a visual overview of these estimators. We also include examples demonstrating how EGM accommodates diffusion, flow, and jump processes.[2]

---

[2]We include the derivation and example in Appendix B for mixed state space.

## 3.1 Marginal generator estimation via self-normalized importance sampling

**Problem setup.** We consider the sampling problem from a density[3] $p_1(x) = \tilde{p}_1(x)/Z$ given only the unnormalized density $\tilde{p}_1(x) = \exp(-\mathcal{E}_1(x))$ with intractable partition function $Z = \int \tilde{p}_1(dx)$. Following GM, our goal is to train a neural network $F_t^\theta$ to approximate the parametrization $F_t$ of the marginal generator $\mathcal{L}_t$ in Equation (3).

**SNIS estimation of the generator.** First, consider the following importance sampling estimator:

$$\mathcal{L}_t f(x) = \mathbb{E}_{x_1 \sim q_{1|t}(\cdot|x)} \left[ w(x, x_1) \mathcal{L}_{t|1}^{x_1} f(x) \right], \quad w(x, x_1) := \frac{p_{1|t}(x_1|x)}{q_{1|t}(x_1|x)} = \frac{\tilde{p}_1(x_1) p_{t|1}(x|x_1)}{Z p_t(x) q_{1|t}(x_1|x)}, \tag{7}$$

where $p_{1|t}$ is the target posterior and $q_{1|t}$ is a proposal kernel. We avoid computing intractable $Z p_t(x)$ for the importance weights by using SNIS scheme:

$$\tilde{w}(x, x_1) := \frac{\tilde{p}_1(x_1) p_{t|1}(x|x_1)}{q_{1|t}(x_1|x)}, \quad w(x, x_1) = \frac{\tilde{w}(x, x_1)}{Z p_t(x)} = \frac{\tilde{w}(x, x_1)}{\mathbb{E}_{x_1 \sim q_{1|t}(\cdot|x)}[\tilde{w}(x, x_1)]}, \tag{8}$$

where $\tilde{w}$ is the unnormalized importance weight that is tractable to compute. In this way, one can compute the self-normalized weight without knowing the normalization constant $Z p_t(x)$. Furthermore, when we use the same samples for estimating the generator and normalization constant, we obtain a low-variance estimator at the cost of inducing a bias.

**Energy-based generator matching (EGM).** The linear parametrization of the generator admits the importance sampling expression $F_t(x) = \mathbb{E}_{x_1 \sim q_{1|t}(\cdot|x)} \left[ w(x, x_1) F_{t|1}^{x_1}(x) \right]$. Then the loss is:

$$L_{\text{EGM}}(\theta) = \mathbb{E}_{t \sim \text{Unif}, \, x_t \sim p_t^{\text{ref}}} \left[ D\left( \hat{F}_t(x_t), F_t^\theta(x_t) \right) \right], \quad \hat{F}_t(x) = \frac{\sum_i \tilde{w}(x, x_1^{(i)}) F_{t|1}^{x_1^{(i)}}(x)}{\sum_i \tilde{w}(x, x_1^{(i)})}, \tag{9}$$

where $x_1^{(i)}$ are sampled from the proposal $q_{1|t}(\cdot|x)$. We implement $\hat{F}_t$ using the `LogSumExp` trick, which provides a numerically stable estimator even in the low-density region.

**Example 1: Conditional optimal-transport (OT) path.** EGM can be applied to train a neural flow sampler driven by the conditional OT path [25]. In the flow model[4], the conditional probability path and its conditional velocity field are expressed by $p_{t|1}(x|x_1) = \mathcal{N}(x; tx_1, (1-t)^2 \text{Id})$ and $u_{t|1}(x|x_1) = \frac{x_1 - x}{1-t}$ for $x, x_1 \in \mathbb{R}^d$. Choosing the proposal $q_{1|t}(x_1|x) \propto p_{t|1}(x|x_1)$, one obtains,

$$q_{1|t}(x_1|x) = \mathcal{N}\left( x_1; \frac{x}{t}, \frac{(1-t)^2}{t^2} I \right), \quad \hat{u}_t(x) = \frac{\sum_i \tilde{p}_1(x_1^{(i)}) u_{t|1}(x|x_1^{(i)})}{\sum_i \tilde{p}_1(x_1^{(i)})}. \tag{10}$$

**Example 2: Discrete masked diffusion.** EGM also supports masked diffusion path[5], which has recently gained popularity in language modeling. In the masked diffusion path[2], the conditional probability path and the conditional transition rate matrix are expressed as $p_{t|1}(x|x_1) = \kappa_t \delta_{x_1}(x) + (1 - \kappa_t)\delta_M(x)$ and $u_{t|1}(y, x|x_1) = \frac{\dot{\kappa}_t}{1 - \kappa_t}(\delta_{x_1}(y) - \delta_x(y))$ where $M \in S$ is the mask token, $\kappa_t : [0, 1] \to \mathbb{R}$ is a schedule, and $\delta_x$ is Dirac delta distribution at $x$. In this setting, the proposal and the estimator are,

$$q_{1|t}(x_1|x) = \begin{cases} \text{Unif}(x_1; S - M) & (x = M) \\ \delta_x(x_1) & (x \neq M) \end{cases}, \quad \hat{u}_t(y, x) = \frac{\sum_i \tilde{p}_1(x_1^{(i)}) u_{t|1}(y, x|x_1^{(i)})}{\sum_i \tilde{p}_1(x_1^{(i)})}. \tag{11}$$

## 3.2 Bootstrapping tricks for low-variance estimation

The quality of the estimator in Equation (9) depends critically on the proposal distribution $q_{1|t}$. In the ideal case, the proposal $q_{1|t}$ that matches the posterior $p_{1|t}$ ensures that EGM is unbiased and aligns with GM. However, the simple choice of $q_{1|t}(x_1|x_t) \propto p_{t|1}(x_t|x_1)$ leads to a mismatch with the true posterior $p_{1|t}$, resulting in high variance in the importance weight.

---

[3]Though density may not be defined in general, it is given a priori in the sampling problem.

[4]For a detailed description of flow model and masked diffusion path, see Appendix C.

[5]Discrete masked diffusion is a discrete jump process which has transition rate matrix $u_t(y, x)$.

For a nearby future time step $r > t$, the posterior $p_{r|t}(x_r|x_t)$ is similar to the backward transition kernel $p_{t|r}(x_t|x_r)$ because $\frac{p_{r|t}(x_r|x_t)}{p_{t|r}(x_t|x_r)} = \frac{p_r(x_r)}{p_t(x_t)} \approx 1$ for continuous densities. Thus, a simple proposal $q_{r|t}(x_r|x_t) \propto p_{t|r}(x_t|x_r)$ can effectively match the posterior $p_{r|t}(x_r|x_t)$, reducing the variance of importance weights. To exploit this, we derive a generalized marginalization trick using the intermediate state $x_r$.

**Backward transition kernel with marginal consistency.** To derive bootstrapping, we need to choose a backward transition kernel $p_{t|r}$ which satisfies the marginal consistency described by the Chapman-Kolmogorov equation:

$$p_{t|1}(\cdot|x_1) = \int p_{t|r}(\cdot|x_r)p_{r|1}(dx_r|x_1). \tag{12}$$

Not every $p_{t|r}$ with marginal consistency is tractable. For instance, in the flow model, $p_{t|r}$ is deterministic, making density evaluation infeasible. We give examples of tractable backward transition kernels satisfying our condition for diffusion, flow, and masked-diffusion paths in Appendix C.

**Bootstrapped marginalization trick.** Now, we generalize Equation (3) and show that the marginal generator $\mathcal{L}_t$ can be expressed as marginalization over generators $\mathcal{L}_{t|r}^{x_r}$ for probability paths $(p_{t|r}(dx|x_r))_{0 \leq t \leq r}$ conditioned on time $r$, instead of the conditional generators $\mathcal{L}_{t|1}^{x_1}$. To this end, we define marginal consistency of conditional generators as follows:

$$\mathbb{E}_{x_r \sim p_{r|1,t}(\cdot|x_1,x)}[\mathcal{L}_{t|r}^{x_r}f(x)] = \mathcal{L}_{t|1}^{x_1}f(x). \tag{13}$$

With generators $\mathcal{L}_{t|r}^{x_r}$ conditioned on time $r$ satisfying the consistency, we express the marginal generator $\mathcal{L}_t$. We provide the corresponding proof in Appendix A.

**Theorem 1.** *Let $\mathcal{L}_{t|r}^{x_r}$ denote the conditional generator for conditional probability path $p_{t|r}(\cdot|x_r)$ for $0 \leq t < r \leq 1$. If the backward transition kernels $p_{t|r}$ satisfy the Equation (12) and the conditional generators satisfy Equation (13), then the marginal generator can be expressed as follows, regardless of $r$:*

$$\mathcal{L}_t f(x) = \mathbb{E}_{x_r \sim p_{r|t}(\cdot|x)}[\mathcal{L}_{t|r}^{x_r}f(x)], \tag{14}$$

*where $p_{r|t}(dx_r|x)$ is the posterior distribution (i.e., the conditional distribution over intermediate state $x_r$ given an observation $x$ at time $t$).*

**Bootstrapped SNIS estimation of the generator.** Using a proposal distribution $q_{r|t}(\cdot|x)$, the marginal generator can be estimated as follows:

$$\mathcal{L}_t f(x) = \mathbb{E}_{x_r \sim q_{r|t}(\cdot|x)}[w(x,x_r)\mathcal{L}_{t|r}^{x_r}f(x)], \quad w(x,x_r) = \frac{p_{r|t}(x_r|x)}{q_{r|t}(x_r|x)} = \frac{p_r(x_r)p_{t|r}(x|x_r)}{p_t(x)q_{r|t}(x_r|x)}. \tag{15}$$

Similar to Section 3.1, we also consider the estimator based on the self-normalized importance sampling scheme to reduce the variance of the IS estimator. With a simple choice of $q_{r|t}(x_r|x_t) \propto p_{t|r}(x_t|x_r)$, unnormalized importance weight becomes:

$$\tilde{w}(x,x_r) = \frac{\tilde{p}_r(x_r)}{\mathbb{E}_{x_r \sim q_{r|t}(\cdot|x)}[\tilde{p}_r(x_r)]}, \quad \tilde{p}_r(x_r) := \int p_{r|1}(x_r|x_1)\tilde{p}_1(dx_1). \tag{16}$$

Then we obtain the following marginal estimators:

$$\hat{\mathcal{L}}_t f(x) = \frac{\sum_i \tilde{w}(x,x_r^{(i)})\mathcal{L}_{t|r}^{x_r^{(i)}}f(x)}{\sum_i \tilde{w}(x,x_r^{(i)})}, \qquad \hat{F}_t(x) = \frac{\sum_i \tilde{w}(x,x_r^{(i)})F_{t|r}^{x_r^{(i)}}(x)}{\sum_i \tilde{w}(x,x_r^{(i)})}, \tag{17}$$

where $x_r^{(i)}$ are sampled from the proposal $q_{r|t}(\cdot|x)$ and $F_{t|r}^{x_r}$ is the parametrization of $\mathcal{L}_{t|r}^{x_r}$.

**Intermediate energy estimator.** Note that the unnormalized density $\tilde{p}_r$ is intractable, unlike $\tilde{p}_1$ in Equation (9). Therfore, we train a surrogate $\tilde{p}_r^\phi(x)$ to approximate the unnormalized density of the intermediate state $x_r$. This surrogate can be learned using the following estimator:

$$\tilde{p}_r(x_r) := \int p_{r|1}(x_r|x_1)\tilde{p}_1(x_1)dx_1 = Z_{1|r}(x_r)\mathbb{E}_{x_1 \sim q_{1|r}(\cdot|x_r)}[\tilde{p}_1(x_1)], \tag{18}$$

---

**Algorithm 1** Iterated training with EGM loss with bootstrapping

---

**Require:** Network $F_t^\theta, \mathcal{E}_t^\phi$, replay buffer $\mathcal{B} \leftarrow \emptyset$, bootstrapping step size $\epsilon$ and batch size $b$.
1: **while** Outer-loop **do**
2:     Sample $\{x_1\}_{i=1}^b$ from the simulation of the current sampler $F_t^\theta$ and set $\mathcal{B} \leftarrow \mathcal{B} \cup \{x_1\}_{i=1}^b$.
3:     **while** Inner-loop **do**
4:         **if** bootstrapping **then**
5:             Sample $t \sim \mathrm{Unif}[0,1]$, $r \leftarrow \min(t+\epsilon, 1)$, $x_1 \sim \mathcal{B}$ and $x_t \sim p_{t|1}(\cdot|x_1)$.
6:             Update $\phi$ to minimize $L_{\text{NEM}}$ as defined in Equation (20).
7:             Compute the bootstrapped estimator $\hat{F}_t(x; \phi, r)$ with the proposed sample $x_r^{(i)} \sim q_{r|t}(\cdot|x_t)$.
8:             Update $\theta$ to minimize $L_{\text{EGM-BS}}$ as defined in Equation (21).
9:         **else**
10:           Compute the SNIS estimator $\hat{F}_t(x)$ with the proposed sample $x_1^{(i)} \sim q_{1|t}(\cdot|x_t)$.
11:           Update $\theta$ to minimize $L_{\text{EGM}}$ as defined in Equation (9).
12:         **end if**
13:     **end while**
14: **end while**

---

where $Z_{1|r}(x_r)$ is the normalization constant of the proposal $q_{1|r}(x_1|x_r) \propto p_{r|1}(x_r|x_1)$. The estimator is unbiased, but exhibits high variance. Therefore, we learn the energy of intermediate state, $\mathcal{E}_r(x_r) := -\log \tilde{p}_r(x_r)$ that can be expressed using the target energy function $\mathcal{E}_1(x)$:

$$\mathcal{E}_r(x_r) = -\log \mathbb{E}_{x_1 \sim q_{1|r}(\cdot|x_r)}[\exp(-\mathcal{E}_1(x_1))] - \log Z_{1|r}(x_r), \tag{19}$$

To learn the surrogate, we train on a general version of noised energy matching (NEM) [30] objective:

$$L_{\text{NEM}}(\phi) = \mathbb{E}_{x_r \sim p_r^{\text{ref}}}[\|\mathcal{E}_r^\phi(x_r) - \hat{\mathcal{E}}_r(x_r)\|_2^2], \ \ \hat{\mathcal{E}}_r(x_r) = -\log \frac{1}{K} \sum_i \exp(-\mathcal{E}_1(x_1^{(i)})) - \log Z_{1|r}(x_r), \tag{20}$$

where $x_1^{(1)}, \ldots, x_1^{(K)}$ is sampled from $q_{1|t}$. Then our final loss from Equation (9) becomes:

$$L_{\text{EGM-BS}}(\theta; \phi) = \mathbb{E}_{x_t \sim p_t^{\text{ref}}}\left[D\big(\hat{F}_t(x_t; \phi), F_t^\theta(x_t)\big)\right], \ \ \hat{F}_t(x_t; \phi, r) = \frac{\sum_i \exp(-\mathcal{E}_r^\phi(x_r^{(i)})) F_{t|r}^{x_r^{(i)}}(x_t)}{\sum_i \exp(-\mathcal{E}_r^\phi(x_r^{(i)}))}, \tag{21}$$

where $x_r^{(i)}$ is sampled from the proposal $q_{r|t}(\cdot|x)$.

**Example 1: Conditional OT path.** To apply bootstrapping with conditional OT path, we show that the backward transition kernel $p_{t|r}(x_t|x_r) = \mathcal{N}\left(x_t; \frac{t}{r}x_r, \sigma_t I\right)$ and conditional velocity $u_{t|r}(x_t|x_r) = \frac{1}{r}x_r + \frac{\dot{\sigma}_t}{2\sigma_t}\left(x_t - \frac{t}{r}x_r\right)$ satisfy backward Kolmogorov equation with above proposed conditional probability path $p_{t|1}$ where $\sigma_t = (1-t)^2 - \frac{t^2}{r^2}(1-r)^2$. Choosing the proposal $q_{r|t}(x_r|x_t) \propto p_{t|r}(x_t|x_r)$, one obtains,

$$q_{r|t}(x_r|x_t) = \mathcal{N}\left(x_t; \frac{r}{t}x_t, \frac{r^2}{t^2}\sigma_t I\right), \quad \hat{u}_t(x_t) = \frac{\sum_i \tilde{p}_r(x_r^{(i)}) u_{t|r}(x_t|x_r)}{\sum_i \tilde{p}_r(x_r^{(i)})}. \tag{22}$$

We check that the transition kernel $p_{t|r}$ satisfies the assumption we proposed in Appendix C.

**Example 2: Discrete masked diffusion.** For masked-diffusion path, the backward transition kernel and conditional transition rate matrix are given as $p_{t|r}(x_t|x_r) = \frac{\kappa_t}{\kappa_r}\delta_{x_r}(x_t) + \frac{\kappa_r - \kappa_t}{\kappa_r}\delta_M(x_t)$ and $u_{t|r}(y, x_t|x_r) = \frac{\dot{\kappa}_t}{\kappa_r - \kappa_t}(\delta_{x_r}(y) - \delta_{x_t}(y))$. Then, the proposal and the estimator are[6],

$$q_{r|t}(x_r|x_t = M) = \frac{\kappa_t}{\kappa_r}\delta_M(x_r) + \frac{\kappa_r - \kappa_t}{\kappa_r}, \quad \hat{u}_t(y, x) = \frac{\sum_i \tilde{p}_r(x_r^{(i)}) u_{t|r}(y, x|x_r^{(i)})}{\sum_i \tilde{p}_r(x_r^{(i)})}. \tag{23}$$

### 3.3 Training details

We now describe the training algorithm for EGM, with the full procedure provided in Algorithm 1.

---

[6]Note that $q_{r|t}(x_r|x_t \neq M) = \delta_{x_t}(x_r)$, the token flipped to the data token does not change thereafter.

**Bi-level training scheme.** To choose a reference distribution $p_t^{\text{ref}}$ that is close to $p_t$, we use hierarchical sampling procedure $x_1 \sim \mathcal{B}$, $x_t \sim p_{t|1}(\cdot|x_1)$ with replay-buffer $\mathcal{B}$ that approximates $p_1$. This leads to the following form of EGM objective:

$$L_{\text{EGM}}(\theta) = \mathbb{E}_{t\sim\text{Unif}, x_1\sim\mathcal{B}, x_t\sim p_{t|1}(\cdot|x_1)}\big[D\big(\hat{F}_t(x_t), F_t^\theta(x_t)\big)\big]. \qquad (24)$$

When the buffer remains close to $p_1$, our loss approximates the generator matching loss in Equation (4). We achieve this by continuously updating the buffer, using the bi-level scheme introduced in Akhound-Sadegh et al. [1]. The bi-level scheme alternates between an outer loop and an inner loop, where in the outer loop the buffer $\mathcal{B}$ is improved by drawing samples from the current sampler $F_t^\theta$. For the inner loop, the sampler $F_t^\theta$ is trained to minimize the EGM loss with $\mathcal{B}$ held fixed. Because the sampler is updated in the inner loop, the samples collected in the subsequent outer loop reflect its improved performance, thus progressively improving the buffer.

**Forward-looking parametrization for masked diffusion.** We also introduce a trick to further boost the training of masked diffusion samplers for graphical models via inductive bias on the estimator $\mathcal{E}_t^\phi$. To this end, consider a graphical model defined on graph $G = (V, E)$ with vertices $V$ and edges $E$. It defines a distribution of vertex-wise variables $x = \{x_i\}_{i\in V}$ using an energy function $\mathcal{E}$ expressed as a product of edge-wise potentials $\{\psi_{i,j}\}_{\{i,j\}\in E}$, i.e., $p_1(x) \propto \exp\big(-\mathcal{E}(x)\big) = \prod_{\{i,j\}\in E} \psi_{i,j}(x^i, x^j)$ where we let $x^i$ denote the variable associated with vertex $i \in V$.

For these models, we can express the intermediate distribution $p_t(x_t)$ using fixed configurations, since once a token is converted from a mask to a data token, it remains fixed thereafter. To this end, let $I_t$ denote the set of edges with deterministic value at time $t$, and the intermediate distribution can be expressed as follows:

$$p_t(x_t) = \prod_{\{i,j\}\in I_t} \psi_{i,j}\big(x_t^i, x_t^j\big) \int p_t(x_t|x_1) \prod_{\{i,j\}\in E\setminus I_t} \psi_{i,j}\big(x_1^i, x_1^j\big)\, dx_1. \qquad (25)$$

Hence, it suffices to estimate the contributions of the undetermined region, which leads to our parameterization $\mathcal{E}_t^\phi$ as: $\mathcal{E}_t^\phi(x_t) = \text{NN}_\phi(x_t, t) - \sum_{(i,j)\in I_t} \log \psi_{i,j}(x_t^i, x_t^j)$.

# 4 Experiments

In prior work, research has concentrated mainly on diffusion-based samplers, leaving purely discrete and multimodal settings underexplored. To fill this gap, we evaluate EGM on purely discrete and joint discrete–continuous tasks. Specifically, we employ jump-based neural samplers for purely discrete tasks and combine discrete jumps with continuous flow to tackle multimodal tasks. These experiments demonstrate EGM's performance and versatility.

Our primary evaluation metric is the energy-$\mathcal{W}_1$ ($\mathcal{E}$-$\mathcal{W}_1$) distance, consistently applied across tasks and complemented by qualitative visualizations. Since no jump-based or hybrid discrete–continuous neural sampler baselines exist, we include a traditional Gibbs sampler [16] as our competitive baseline. Although Gibbs sampling is guaranteed to converge given sufficiently many iterations, we use four parallel chains with 6000 steps each to reflect a realistic computational budget for the per-sample cost and to yield reasonably competitive results. Detailed descriptions of the energy functions, evaluation metrics, and experimental protocols are provided in Appendix D.

## 4.1 Discrete EGM on the 2D Ising Model

We assess the performance and scalability of EGM using the two-dimensional Ising model, varying both grid dimension and temperature. The Ising model is a canonical benchmark in probabilistic inference and sampling, with well-established ground-truth samples facilitating robust evaluation. The Ising model, whose complexity can be tuned via temperature and grid size, serves as an ideal testbed for our jump-based sampler.

We report the quantitative results in Table 2, alongside comparisons to the traditional parallel-Gibbs sampler as a baseline. Given the absence of metric structure in the discrete Ising model, we adopt the $\mathcal{W}_1$ distance over energy and average magnetization $M = \sum_i x_i$ as metrics. EGM consistently matches or exceeds baseline performance across metrics. Especially, bootstrapping improves the performance even in high-dimensional settings near critical temperatures $\beta = 0.4$.

Table 2: Performance of *EGM* on Ising model in terms of energy $\mathcal{W}_1$ ($\mathcal{E}$-$\mathcal{W}_1$) and magnetization $\mathcal{W}_1$ ($M$-$\mathcal{W}_1$). For reference, we report the metric of the Gibbs sampler. Each sampler is evaluated with three random seeds, and we report the mean $\pm$ standard deviation for each metric. Results shown in **bold** denote the best result in each column.

| Energy → | 5 x 5 Ising ($d = 25$) | | | | 10 x 10 Ising ($d = 100$) | | | |
|---|---|---|---|---|---|---|---|---|
| Parameters → | $\beta = 0.2, J = 1.0$ | | $\beta = 0.4, J = 1.0$ | | $\beta = 0.2, J = 1.0$ | | $\beta = 0.4, J = 1.0$ | |
| Algorithm ↓ | $\mathcal{E}$-$\mathcal{W}_1 \downarrow$ | $M$-$\mathcal{W}_1 \downarrow$ | $\mathcal{E}$-$\mathcal{W}_1 \downarrow$ | $M$-$\mathcal{W}_1 \downarrow$ | $\mathcal{E}$-$\mathcal{W}_1 \downarrow$ | $M$-$\mathcal{W}_1 \downarrow$ | $\mathcal{E}$-$\mathcal{W}_1 \downarrow$ | $M$-$\mathcal{W}_1 \downarrow$ |
| **Gibbs** | **0.10**±0.02 | 0.06±0.02 | 0.71±0.43 | 0.23±0.18 | 0.53±0.04 | 0.04±0.01 | 5.29±1.95 | 0.29±0.11 |
| **EGM (ours)** | 0.20±0.06 | **0.02**±0.01 | 3.73±0.39 | 0.24±0.02 | 0.84±0.07 | **0.02**±0.00 | 19.94±0.69 | 0.36±0.01 |
| **+Bootstrapping (ours)** | **0.10**±0.06 | **0.02**±0.01 | **0.60**±0.12 | **0.04**±0.01 | **0.39**±0.34 | 0.06±0.07 | **2.51**±0.16 | **0.24**±0.01 |

Figure 2: Comparison of energy (top) and magnetization (bottom) histograms for ground-truth samples and various sampling methods.

Qualitative evaluations presented in Figure 2 depict energy and average magnetization histograms for the Ising model with low temperature $\beta = 0.4$, showing EGM's ability to accurately capture the true energy distribution.

## 4.2 Multimodal EGM on GB-RBM, JointDW4 and JointMoG

We validate EGM on joint discrete–continuous sampling tasks using three synthetic benchmarks: Gaussian-Bernoulli restricted Boltzmann machine (GB-RBM) [9], double-well potential with type-dependent interactions (JointDW4), and discrete-continuous joint mixture of Gaussians (JointMoG). We choose GB-RBM to show an application of sampling from the energy-based model. JointDW4 mimics a simplified molecular sequence-structure co-generation problem. JointMoG is a mixed-state extension of MoG that is common in diffusion-sampler benchmarks. For all the experiments, we adopt conditional-OT or variance exploding paths as a conditional probability path for the continuous flow sampler, and a masked diffusion path for the discrete jump sampler.

Quantitative outcomes are summarized in Table 3. For GB-RBM, we also use $x$-$\mathcal{W}_2$ for the first two continuous dimensions. Qualitative results for GB-RBM and JointMoG in Figure 3 visually demonstrate EGM's capacity to capture all distinct modes accurately. In Figure 4, we observe that EGM (with bootstrapping) correctly models the true energy distribution in JointMoG compared to the Gibbs sampler. Bootstrapping consistently improves performance in capturing multiple modes of the distributions.

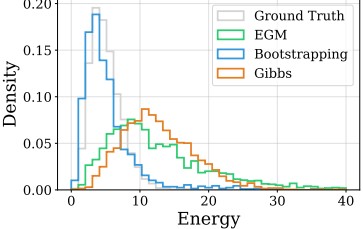

Figure 4: Energy histograms on the samples from multiple samplers vs. ground truth of JointMOG.

## 5 Related Works

**Diffusion samplers.** Diffusion samplers cast sampling as a stochastic control or denoising problem. Early work, such as the path integral sampler (PIS) [42], required on-policy SDE simulations per update, leading to high computational costs. Subsequent variants explored different probability

Table 3: Performance of *EGM* on multimodal task. For reference, we report the metric of the Gibbs sampler. Each sampler is evaluated with three random seeds, and we report the mean $\pm$ standard deviation for each metric. Results shown in **bold** denote the best result in each column.

| Energy $\rightarrow$ | GB-RBM ($d = 5$) | | JointDW4 ($d = 12$) | JointMoG ($d = 20$) |
|---|---|---|---|---|
| Algorithm $\downarrow$ | $\mathcal{E}$-$\mathcal{W}_1 \downarrow$ | $x$-$\mathcal{W}_2 \downarrow$ | $\mathcal{E}$-$\mathcal{W}_1 \downarrow$ | $\mathcal{E}$-$\mathcal{W}_1 \downarrow$ |
| **Gibbs** | $0.77_{\pm 0.03}$ | $5.12_{\pm 0.21}$ | $4.80_{\pm 0.02}$ | $7.76_{\pm 0.02}$ |
| **EGM (ours)** | $0.33_{\pm 0.02}$ | $0.63_{\pm 0.10}$ | $2.45_{\pm 0.07}$ | $8.83_{\pm 0.12}$ |
| **+Bootstrapping (ours)** | $\mathbf{0.20}_{\pm 0.15}$ | $\mathbf{0.61}_{\pm 0.09}$ | $\mathbf{1.65}_{\pm 0.96}$ | $\mathbf{1.20}_{\pm 0.32}$ |

(a) Gibbs    (b) EGM    (c) EGM + BS    (d) Gibbs    (e) EGM    (f) EGM + BS

Figure 3: Sample plots of GB-RBM (a-c) and JointMoG (d-f). Samples are projected onto the first two continuous dimensions. BS stands for bootstrapping. Contour lines represent the target distribution, and colored points indicate samples from each method.

paths [38, 5, 39] and divergences [39, 31], yet most still rely on solving or simulating SDEs to compute their objectives. To reduce this burden, off-policy methods like iDEM [1], iEFM [40], and BNEM [30] estimate the marginal score (or flow) directly and eliminate inner-loop simulation; PINN-based approaches such as NETS [2] similarly train samplers without rollouts. While these advances improve efficiency in continuous spaces, discrete or jump process samplers remain underexplored. Recently, LEAPS [20] introduced a proactive importance sampling scheme for discrete state spaces using locally equivariant networks. However, it requires many energy evaluations at inference since it uses escorted transport.

**Generative flow networks (GFNs).** GFNs [3, 22] formulate sampling as a sequential decision process: a forward policy constructs an object step by step, and a backward policy enforces consistency so that trajectories terminate with probability proportional to a target reward (or unnormalized density). Training objectives like trajectory balance [26] and detailed balance [4] enable off-policy updates. Furthermore, Falet et al. [12] propose the sampler for a sparse graphical model with a local objective similar to our forward-looking parametrization. Though diffusion sampler is a special case of a continuous-state continuous-time GFNs [41, 6], it remains an interesting question that training a CTMP on a general state space, as ours, can be unified under the GFNs framework.

**Relation to iDEM and BNEM.** In EGM, choosing $q_{1|t}(x_1|x) \propto p_{t|1}(x|x_1)$ yields simple weights $\tilde{w}(x, x_1) = \tilde{p}_1(x_1) = \exp(-\mathcal{E}(x_1))$. In the VE path on continuous space where target score identity [11] is available, the integrand can be replaced with the target score instead of the conditional score. This yields the same estimator used in iDEM [1] and NEM [30]. Yet, it remains an open question whether it's possible to derive the target score identity on a general state space. Recently, Zhang et al. [43] introduce the target score identity on discrete states with neighborhood structure, which might improve the accuracy of our estimator on discrete space when combined.

# 6 Conclusion

We introduce energy-based generator matching (EGM), a training method for neural samplers on general state spaces that directly estimates the marginal generator of a continuous-time Markov process (CTMP). EGM is the first neural-sampler framework to handle discrete, continuous, and hybrid multimodal distributions via a principled generator-matching objective. We also propose a bootstrapping scheme using intermediate energy estimates to reduce variance in importance weights. We empirically validate EGM on both high-dimensional discrete systems and hybrid discrete–continuous domains. Our results show that EGM, especially with bootstrapping, performs competitively when sampling from complex distributions with multiple modes. This work opens new avenues for training expressive neural samplers beyond diffusion-based models. Future work may explore unbiased and low-variance estimation of the generator, learned proposal for better estimation, theoretical

connections to physics-informed neural network (PINN) objectives, and unifying the training of general CTMP into the GFNs framework.

## Acknowledgments and Disclosure of Funding

This work was partly supported by Institute for Information & communications Technology Planning & Evaluation(IITP) grant funded by the Korea government(MSIT) (RS-2019-II190075, Artificial Intelligence Graduate School Support Program(KAIST)), National Research Foundation of Korea(NRF) grant funded by the Ministry of Science and ICT(MSIT) (No. RS-2022-NR072184), GRDC(Global Research Development Center) Cooperative Hub Program through the National Research Foundation of Korea(NRF) grant funded by the Ministry of Science and ICT(MSIT) (No. RS-2024-00436165), and the Institute of Information & Communications Technology Planning & Evaluation(IITP) grant funded by the Korea government(MSIT) (RS-2025-02304967, AI Star Fellowship(KAIST)). Minsu Kim acknowledges funding from KAIST Jang Yeong Sil Fellowship.

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

# A Proofs

## A.1 Importance sampling for the generator estimation

This section provides a detailed derivation of the importance sampling estimator for the generator and its parametrization presented in Equation (9).

**Existence of densities.** Consider the measurable space $(S, \Sigma)$ with a $\Sigma$-measurable reference measure $\nu$ (e.g., counting measure if $S$ is discrete and Lebesgue measure if $S = \mathbb{R}^d$), as introduced in the Section 2. Since we focus on a sampling problem, the target distribution $p_1$ admits a $\nu$-density $\frac{dp_1}{d\nu} : S \to \mathbb{R}_{\geq 0}$. Assume that the joint probability measure $p_{t,1}$ is absolutely continuous with respect to the product measure $\nu \otimes \nu$. Then, all related probability measures $p_t, p_{t|1}, p_{1|t}$ admits corresponding $\nu$-densities, expressed as:

$$p_t(dx_t) = \int p_{t,1}(dx_t, dx_1) \tag{26}$$

$$= \int p_{t,1}(x_t, x_1)(\nu \otimes \nu)(dx_t, dx_1) \tag{27}$$

$$= \underbrace{\left( \int p_{t,1}(x_t, x_1)\nu(dx_1) \right)}_{p_t(x_t)} \nu(dx_t), \tag{28}$$

$$p_{t,1}(dx_t, dx_1) = \int p_{t,1}(x_t, x_1)(\nu \otimes \nu)(dx_t, dx_1) \tag{29}$$

$$= \int \frac{p_{t,1}(x_t, x_1)}{p_t(x_t)} p_t(x_t)\nu(dx_t)\nu(dx_1) \tag{30}$$

$$= \int \underbrace{\frac{p_{t,1}(x_t, x_1)}{p_t(x_t)}}_{=p_{1|t}(x_1|x_t)} \nu(dx_1)p_t(dx_t), \tag{31}$$

$$p_{t,1}(dx_t, dx_1) = \int p_{t,1}(x_t, x_1)(\nu \otimes \nu)(dx_t, dx_1) \tag{32}$$

$$= \int \frac{p_{t,1}(x_t, x_1)}{p_1(x_1)} p_1(x_1)\nu(dx_1)\nu(dx_t) \tag{33}$$

$$= \int \underbrace{\frac{p_{t,1}(x_t, x_1)}{p_1(x_1)}}_{=p_{t|1}(x_t|x_1)} \nu(dx_t)p_1(dx_1), \tag{34}$$

where $p_{t,1}(x_t, x_1)$ denotes the density of the joint probability measure.

**SNIS estimation of the generator.** We introduce a proposal distribution $q_{1|t} : \Sigma \times S \to \mathbb{R}_{\geq 0}$, satisfying absolute continuity conditions: $p_{1|t}(\cdot|x) \ll q_{1|t}(\cdot|x)$, $q_{1|t}(\cdot|x) \ll \nu$ and $\nu \ll q_{1|t}(\cdot|x)$ for all $x \in S$. Using the marginalization trick Equation (3) and the Radon-Nikodym theorem [23, Chapter 7], we have:

$$\mathcal{L}_t f(x) = \mathbb{E}_{x_1 \sim p_{1|t}(\cdot|x)}[\mathcal{L}_{t|1}^{x_1} f(x)] \tag{35}$$

$$= \mathbb{E}_{x_1 \sim q_{1|t}(\cdot|x)} \left[ \frac{dp_{1|t}}{dq_{1|t}}(x_1|x)\mathcal{L}_{t|1}^{x_1} f(x) \right]. \tag{36}$$

Since the Bayes' rule holds for the $\nu$-density, i.e., $p_{1|t}(x_1|x_t) = \frac{p_{t|1}(x_t|x_1)p_1(x_1)}{p_t(x_t)}$, we obtain:

$$\mathcal{L}_t f(x_t) = \mathbb{E}_{x_1 \sim q_{1|t}(\cdot|x_t)}\left[\frac{dp_{1|t}}{dq_{1|t}}(x_1|x_t)\mathcal{L}_{t|1}^{x_1}f(x_t)\right] \tag{37}$$

$$= \mathbb{E}_{x_1 \sim q_{1|t}(\cdot|x_t)}\left[\frac{dp_{1|t}}{d\nu}(x_1|x_t)\frac{d\nu}{dq_{1|t}}(x_1|x_t)\mathcal{L}_{t|1}^{x_1}f(x_t)\right] \tag{38}$$

$$= \mathbb{E}_{x_1 \sim q_{1|t}(\cdot|x_t)}\left[\frac{p_{1|t}(x_1|x_t)}{q_{1|t}(x_1|x_t)}\mathcal{L}_{t|1}^{x_1}f(x_t)\right] \tag{39}$$

$$= \mathbb{E}_{x_1 \sim q_{1|t}(\cdot|x_t)}\left[w(x_t, x_1)\mathcal{L}_{t|1}^{x_1}f(x_t)\right] \tag{40}$$

where $w(x_t, x_1) := \frac{p_{1|t}(x_1|x_t)}{q_{1|t}(x_1|x_t)} = \frac{\tilde{p}_1(x_1)p_{t|1}(x_t|x_1)}{Zp_t(x_t)q_{1|t}(x_1|x_t)}$. We estimate the normalization term $Zp_t(x_t)$ with tractable unnormalized density $\tilde{w}(x_t, x_1) := \frac{\tilde{p}_1(x_1)p_{t|1}(x_t|x_1)}{q_{1|t}(x_1|x_t)}$:

$$\mathbb{E}_{x_1 \sim q_{1|t}(\cdot|x)}[\tilde{w}(x_t, x_1)] = \mathbb{E}_{x_1 \sim q_{1|t}(\cdot|x)}\left[\frac{\tilde{p}_1(x_1)p_{t|1}(x_t|x_1)}{q_{1|t}(x_1|x_t)}\right] \tag{41}$$

$$= \int \tilde{p}_1(x_1)p_{t|1}(x_t|x_1)\nu(dx_1) \tag{42}$$

$$= \int Z\frac{\tilde{p}_1(x_1)}{Z}p_{t|1}(x_t|x_1)\nu(dx_1) \tag{43}$$

$$= Z \int p_1(x_1)p_{t|1}(x_t|x_1)\nu(dx_1) \tag{44}$$

$$= Zp_t(x_t). \tag{45}$$

Thus, we derive the self-normalized importance sampling (SNIS) estimator for the generator:

$$\mathcal{L}_t f(x_t) = \frac{\mathbb{E}_{x_1 \sim q_{1|t}(\cdot|x_t)}\left[\tilde{w}(x_t, x_1)\mathcal{L}_{t|1}^{x_1}f(x_t)\right]}{\mathbb{E}_{x_1 \sim q_{1|t}(\cdot|x_t)}[\tilde{w}(x_t, x_1)]} \tag{46}$$

**SNIS estimation of the parametrization.** Similarly, the SNIS estimator for the parameterization $F_t$ is:

$$F_t(x_t) = \mathbb{E}_{x_1 \sim p_{1|t}(\cdot|x_t)}[F_{t|1}^{x_1}(x_t)] \tag{47}$$

$$= \mathbb{E}_{x_1 \sim q_{1|t}(\cdot|x_t)}\left[\frac{dp_{1|t}}{dq_{1|t}}(x_1|x_t)F_{t|1}^{x_1}(x_t)\right] \tag{48}$$

$$= \mathbb{E}_{x_1 \sim q_{1|t}(\cdot|x_t)}\left[\frac{p_{1|t}(x_1|x_t)}{q_{1|t}(x_1|x_t)}F_{t|1}^{x_1}(x_t)\right] \tag{49}$$

$$= \frac{\mathbb{E}_{x_1 \sim q_{1|t}(\cdot|x_t)}\left[\tilde{w}(x_t, x_1)F_{t|1}^{x_1}(x_t)\right]}{\mathbb{E}_{x_1 \sim q_{1|t}(\cdot|x_t)}[\tilde{w}(x_t, x_1)]} \tag{50}$$

Specifically, the expression above suggests the Monte-Carlo (MC) estimator with $K$ samples $x_1^{(1)}, \ldots, x_1^{(K)} \sim q_{1|t}(\cdot|x)$ as follows:

$$\hat{F}_t(x_t) = \frac{\sum_{i=1}^{K} \tilde{w}(x_t, x_1^{(i)})F_{t|1}^{x_1^{(i)}}(x_t)}{\sum_{i=1}^{K} \tilde{w}(x_t, x_1^{(i)})}. \tag{51}$$

This is a SNIS estimator of the parametrization $F_t(x)$.

## A.2    Proof of Theorem 1

For convenience, we repeat the theorem and its assumptions below.

**Theorem 2** (Restatement of Theorem 1). *Let $\mathcal{L}_{t|r}^{x_r}$ denote the conditional generator for conditional probability path $p_{t|r}(\cdot|x_r)$ for $0 \leq t \leq r \leq 1$. If the backward transition kernels $p_{t|r}$ satisfy the Chapman-Kolmogorov equation,*

$$p_{t|1}(dx_t|x_1) = \int p_{t|r}(dx_t|x_r)p_{r|1}(dx_r|x_1), \tag{52}$$

*and the conditional generators $\mathcal{L}_{t|r}^{x_r}$ satisfy the marginal consistency as follows,*

$$\mathbb{E}_{x_r \sim p_{r|1,t}(\cdot|x_1,x_t)}\left[\mathcal{L}_{t|r}^{x_r}f(x_t)\right] = \mathcal{L}_{t|1}^{x_1}f(x_t). \tag{53}$$

*Then the marginal generator can be expressed as follows, regardless of $r$:*

$$\mathcal{L}_t f(x_t) = \mathbb{E}_{x_r \sim p_{r|t}(\cdot|x_t)}\left[\mathcal{L}_{t|r}^{x_r}f(x_t)\right], \tag{54}$$

*where $p_{r|t}(dx_r|x)$ is the posterior distribution (i.e., the conditional distribution over intermediate state $x_r$ given an observation $x$ at time $t$).*

*Proof.* Define the marginal generator conditioned at time $r$ as $\mathcal{L}_{t;r}f(x) := \mathbb{E}_{x_r \sim p_{r|t}(\cdot|x)}[\mathcal{L}_{t|r}^{x_r}f(x)]$. Although this definition depends explicitly on $r$, we aim to demonstrate its independence from $r$. This invariance is crucial since any dependence on $r$ would result in conflicting objectives at times $t < r_1, r_2$ for distinct $r_1, r_2$. The proof proceeds in two main steps:

1. Verify that the marginal generator $\mathcal{L}_{t;r}$ generates the probability path $(p_t)_{0 \leq t \leq r}$.

2. Show the marginal generator's independence from $r$ (i.e., $\mathcal{L}_{t;r} = \mathcal{L}_t$).

To establish the first step, it suffices to verify that the Kolmogorov Forward Equation (KFE) holds for the probability path $(p_t)_{0 \leq t \leq r}$ and the generator $\mathcal{L}_{t;r}$:

$$\frac{d}{dt}\mathbb{E}_{x_t \sim p_t}[f(x_t)] = \mathbb{E}_{x_t \sim p_t}[\mathcal{L}_{t;r}f(x_t)] \quad \text{for} \quad 0 \leq t \leq r \leq 1. \tag{55}$$

The KFE is satisfied for the conditional probability path $pt|r$ by definition:

$$\frac{d}{dt}\mathbb{E}_{x_t \sim p_{t|r}(\cdot|x_r)}[f(x_t)] = \mathbb{E}_{x_t \sim p_{t|r}(\cdot|x_r)}\left[\mathcal{L}_{t|r}^{x_r}f(x_t)\right], \quad 0 \leq t \leq r \leq 1, x_r \in S. \tag{56}$$

Thus, we have:

$$\mathbb{E}_{x_t \sim p_t}[\mathcal{L}_{t;r}f(x_t)] = \mathbb{E}_{x_t \sim p_t}\mathbb{E}_{x_r \sim p_{r|t}(\cdot|x_t)}\left[\mathcal{L}_{t|r}^{x_r}f(x_t)\right] \tag{57}$$

$$= \mathbb{E}_{x_r \sim p_r}\mathbb{E}_{x_t \sim p_{t|r}(\cdot|x_r)}\left[\mathcal{L}_{t|r}^{x_r}f(x_t)\right] \tag{58}$$

$$= \mathbb{E}_{x_r \sim p_r}\frac{d}{dt}\mathbb{E}_{x_t \sim p_{t|r}(\cdot|x_r)}[f(x_t)] \tag{59}$$

$$= \frac{d}{dt}\mathbb{E}_{x_r \sim p_r}\mathbb{E}_{x_t \sim p_{t|r}(\cdot|x_r)}[f(x_t)] \tag{60}$$

$$= \frac{d}{dt}\mathbb{E}_{x_t \sim p_t}[f(x_t)] \tag{61}$$

Hence, $\mathcal{L}_{t;r}$ indeed generates the probability path $(p_t)_{0 \leq t \leq r}$.

Next, we demonstrate the independence of $\mathcal{L}_{t;r}$ from the choice of $r$:

$$\mathcal{L}_{t;r}f(x_t) = \mathbb{E}_{x_r \sim p_{r|t}(\cdot|x_t)}\left[\mathcal{L}_{t|r}^{x_r}f(x_t)\right] \tag{62}$$

$$= \mathbb{E}_{x_1 \sim p_{1|t}(\cdot|x_t)}\mathbb{E}_{x_r \sim p_{r|t,1}(\cdot|x_t,x_1)}\left[\mathcal{L}_{t|r}^{x_r}f(x_t)\right] \tag{63}$$

$$= \mathbb{E}_{x_1 \sim p_{1|t}(\cdot|x_t)}\left[\mathcal{L}_{t|1}^{x_1}f(x_t)\right] \tag{64}$$

$$= \mathcal{L}_t f(x_t) \tag{65}$$

where the marginal consistency assumption Equation (13) is applied in the third equality. This concludes the proof. □

## A.3 Derivation of bootstrapping estimator for generator estimation

We derive a bootstrapping estimator for the marginal generator and its parametrization proposed in Equation (21), based on the marginalization trick in Equation (54).

**Bootstrapped SNIS estimation of the generator.** Assume that the backward kernel $p_{t|r}$ admits a $\nu$-density. Then, the posterior $p_{r|t}$ also admits a $\nu$-density. Let the proposal distribution $q_{r|t}:$ $\Sigma \times S \to \mathbb{R}_{\geq 0}$ satisfy $p_{r|t}(\cdot|x) \ll q_{r|t}(\cdot|x)$, $q_{r|t}(\cdot|x) \ll \nu$, and $\nu \ll q_{r|t}(\cdot|x)$ for all $x \in S$. Applying the same change-of-measure trick as before:

$$\mathcal{L}_t f(x_t) = \mathbb{E}_{x_r \sim p_{r|t}(\cdot|x_t)}\left[\mathcal{L}_{t|r}^{x_r} f(x_t)\right] \tag{66}$$

$$= \mathbb{E}_{x_r \sim q_{r|t}(\cdot|x_t)}\left[\frac{dp_{r|t}}{dq_{r|t}}(x_r|x_t)\mathcal{L}_{t|r}^{x_r} f(x_t)\right] \tag{67}$$

$$= \mathbb{E}_{x_r \sim q_{r|t}(\cdot|x_t)}\left[\frac{dp_{r|t}}{d\nu}(x_r|x_t)\frac{d\nu}{dq_{r|t}}(x_r|x_t)\mathcal{L}_{t|r}^{x_r} f(x_t)\right] \tag{68}$$

$$= \mathbb{E}_{x_r \sim q_{r|t}(\cdot|x)}\left[\frac{p_{r|t}(x_r|x_t)}{q_{r|t}(x_r|x_t)}\mathcal{L}_{t|r}^{x_r} f(x_t)\right] \tag{69}$$

$$= \mathbb{E}_{x_r \sim q_{r|t}(\cdot|x)}\left[\frac{p_r(x_r)p_{t|r}(x_t|x_r)}{p_t(x_t)q_{r|t}(x_r|x_t)}\mathcal{L}_{t|r}^{x_r} f(x_t)\right] \tag{70}$$

$$= \mathbb{E}_{x_r \sim q_{r|t}(\cdot|x)}\left[w(x_t, x_r)\mathcal{L}_{t|r}^{x_r} f(x_t)\right], \tag{71}$$

where the importance weight $w(x_t, x_r)$ is given by

$$w(x_t, x_r) := \frac{p_{r|t}(x_r|x_t)}{q_{r|t}(x_r|x_t)} = \frac{\tilde{p}_r(x_r)p_{t|r}(x_t|x_r)}{\tilde{p}_t(x_t)q_{r|t}(x_r|x_t)}. \tag{72}$$

To estimate the unnormalized density $\tilde{p}_t(x_t)$, we define the unnormalized importance weight:

$$\tilde{w}(x_t, x_r) := \frac{\tilde{p}_r(x)p_{t|r}(x_t|x_r)}{q_{r|t}(x_r|x_t)}, \tag{73}$$

and compute:

$$\tilde{p}_t(x_t) = \int p_{t|r}(x_t|x_r)\tilde{p}_r(x_r)\nu(dx_r) \tag{74}$$

$$= \int \frac{p_{t|r}(x_t|x_r)\tilde{p}_r(x_r)}{q_{r|t}(x_r|x_t)}q_{r|t}(x_r|x_t)\nu(dx_r) \tag{75}$$

$$= \mathbb{E}_{x_r \sim q_{r|t}(\cdot|x_t)}\left[\frac{p_{t|r}(x_t|x_r)\tilde{p}_r(x_r)}{q_{r|t}(x_r|x_t)}\right] \tag{76}$$

$$= \mathbb{E}_{x_r \sim q_{r|t}(\cdot|x_t)}\left[\tilde{w}(x_t, x_r)\right]. \tag{77}$$

Thus, the marginal generator can be expressed in SNIS form as:

$$\mathcal{L}_t f(x_t) = \frac{\mathbb{E}_{x_r \sim q_{r|t}(\cdot|x_t)}[\tilde{w}(x_t, x_r)\mathcal{L}_{t|r}^{x_r} f(x_t)]}{\mathbb{E}_{x_r \sim q_{r|t}(\cdot|x_t)}[\tilde{w}(x_t, x_r)]} \tag{78}$$

**Bootstrapped SNIS estimation of the parametrization.** Now we derive a similar expression for the parametrization of the generator. Suppose the conditional generator $\mathcal{L}_{t|r}^{x_r}$ admits a parametrization $F_{t|r}^{x_r}$ such that,

$$\mathcal{L}_{t|r}^{x_r} f(x_t) = \langle \mathcal{K}f(x_t), F_{t|r}^{x_r}(x_t)\rangle, \tag{79}$$

where $\mathcal{K}$ is an operator fixed for each type of Markov processes. From the marginalization trick again:

$$\mathcal{L}_t f(x_t) = \mathbb{E}_{x_r \sim p_{r|t}(\cdot|x_t)} \left[ \mathcal{L}_{t|r}^{x_r} f(x_t) \right] \tag{80}$$

$$= \mathbb{E}_{x_r \sim p_{r|t}(\cdot|x_t)} \left[ \langle \mathcal{K} f(x_t), F_{t|r}^{x_r}(x_t) \rangle \right] \tag{81}$$

$$= \left\langle \mathcal{K} f(x_t), \underbrace{\mathbb{E}_{x_r \sim p_{r|t}(\cdot|x_t)} \left[ F_{t|r}^{x_r}(x_t) \right]}_{:=F_t(x_t)} \right\rangle \tag{82}$$

by linearity of the inner product. Thus, the marginal generator is parametrized by

$$F_t(x_t) = \mathbb{E}_{x_r \sim p_{r|t}(\cdot|x_t)}[F_{t|r}^{x_r}(x_t)]. \tag{83}$$

By applying the same importance sampling trick, we obtain the SNIS estimator:

$$F_t(x_t) = \mathbb{E}_{x_r \sim p_{r|t}(\cdot|x_t)} \left[ F_{t|r}^{x_r}(x_t) \right] \tag{84}$$

$$= \mathbb{E}_{x_r \sim q_{r|t}(\cdot|x_t)} \left[ \frac{dp_{r|t}}{dq_{r|t}}(x_r|x_t) F_{t|r}^{x_r}(x_t) \right] \tag{85}$$

$$= \mathbb{E}_{x_r \sim q_{r|t}(\cdot|x_t)} \left[ \frac{p_{r|t}(x_r|x_t)}{q_{r|t}(x_r|x_t)} F_{t|r}^{x_r}(x_t) \right] \tag{86}$$

$$= \frac{\mathbb{E}_{x_r \sim q_{r|t}(\cdot|x_t)} \left[ \tilde{w}(x_t, x_r) F_{t|r}^{x_r}(x_t) \right]}{\mathbb{E}_{x_r \sim q_{r|t}(\cdot|x_t)} \left[ \tilde{w}(x_t, x_r) \right]}. \tag{87}$$

Specifically, this yields the following Monte Carlo estimator using $K$ samples $x_r^{(1)}, \ldots, x_r^{(K)} \sim q_{r|t}(\cdot|x_t)$:

$$\hat{F}_t(x_t) = \frac{\sum_{i=1}^{K} \tilde{w}(x_t, x_r^{(i)}) F_{t|r}^{x_r^{(i)}}(x_t)}{\sum_{i=1}^{K} \tilde{w}(x_t, x_r^{(i)})} \tag{88}$$

### A.4 Analysis on bias and variance of the IS and bootstrapping estimator

In this section, we investigate the bias and variance of the IS and bootstrapping estimator. The goal of the analysis is to support the bootstrapping estimation with a theoretical argument and demonstrate that it provides a lower variance estimator compared to the IS method. To do so, we first derive asymptotic bounds on bias and variance for IS estimation. Then, we derive the bounds for bootstrapping estimation, assuming that the intermediate energy is well-trained. Under this assumption, we show that bootstrapping reduces estimation variance compared to IS, demonstrating that bootstrapping trades off the estimation variance with the model learning bias.

**Bias and variance of IS estimation.** We show that the SNIS estimator's error decays as $O(1/\sqrt{K})$ and its bais and variance decays as $O(1/K)$, where $K$ is the sample size.

**Proposition 1.** *Consider an unnormalized density $\tilde{p}_1(x_1)$ and a conditional generator $F_{t|1}^{x_1}(x_t)$ evaluated on a sample $x_1 \sim q_{1|t}$. Suppose $\tilde{p}_1(x_1)$ and $\left\| \tilde{p}_1(x_1) F_{t|1}^{x_1}(x_t) \right\|$ are sub-Gaussian. Then there exists a constant $c(x_t)$ such that with probability at least $1 - \delta$,*

$$\left\| \hat{F}_t(x_t) - F_t(x_t) \right\| \leq c(x_t) \sqrt{\frac{\log(2/\delta)}{K}},$$

*where $\hat{F}_t(x_t)$ denotes the SNIS estimator from Equation (9) using $K$ samples $x_1^{(1)}, \ldots, x_1^{(K)} \sim q_{1|t}$:*

$$\hat{F}_t(x_t) = \frac{\sum_{i=1}^{K} \tilde{p}_1(x_1^{(i)}) F_{t|1}^{x_1^{(i)}}(x_t)}{\sum_{i=1}^{K} \tilde{p}_1(x_1^{(i)})}.$$

*Moreover, its bias and variance are expressed as:*

$$Bias[\hat{F}_t] = \frac{1}{K p_t(x_t)^2} \left( -Cov\left[ \tilde{p}_1 F_{t|1}, \tilde{p}_1 \right] + F_t(x_t) Var[\tilde{p}_1] \right) + O\left( \frac{1}{K^2} \right),$$

$$Var[\hat{F}_t] = \frac{4Var[\tilde{p}_1(x_1)]}{p_t^2(x_t)K}(1 + \|F_t(x_t)\|)^2.$$

*Proof.* Let $\hat{A}$ and $\hat{B}$ denote the numerator and the denominator of $\hat{F}_t$, respectively, i.e.,

$$\hat{A} = \frac{1}{K}\sum_{i=1}^{K}\tilde{p}_1(x_1^{(i)})F_{t|1}^{x_1^{(i)}}(x_t), \qquad \hat{B} = \sum_{i=1}^{K}\tilde{p}_1(x_1^{(i)}). \tag{89}$$

We also let $\mathbb{E}[\hat{A}] = A = \tilde{p}_t(x_t)F_t(x_t)$ and $\mathbb{E}[\hat{B}] = B = \tilde{p}_t(x_t)$. By Hoeffding's inequality for sub-Gaussian random variables, there exists a constant $C$ such that

$$\left\|\hat{A} - A\right\| \le C\sqrt{\frac{\log(2/\delta)}{K}}, \qquad \left|\hat{B} - B\right| \le C\sqrt{\frac{\log(2/\delta)}{K}}, \tag{90}$$

with $1 - \delta$ probability. (Here, $C = \sqrt{Var[\tilde{p}_1(x_1)]}$ is a possible choice.)

Since we have bounds for both numerator and denominator of $\hat{F}_t$, we can also bound the error of $\hat{F}_t$ to $F_t$. The result is as follows:

$$\left\|\hat{F}_t(x_t) - F_t(x_t)\right\| = \left\|\frac{\hat{A}}{\hat{B}} - \frac{A}{B}\right\| = \left\|\frac{\hat{A}B - A\hat{B}}{\hat{B}B}\right\| \tag{91}$$

$$\le \frac{\|A\|\left|\hat{B} - B\right| + B\left\|\hat{A} - A\right\|}{\hat{B}B} \tag{92}$$

$$\le \frac{1}{\hat{B}B}Cp_t(x_t)(1 + \|F_t(x_t)\|)\sqrt{\frac{\log(2/\delta)}{K}} \tag{93}$$

$$\le \frac{2C}{p_t(x_t)}(1 + \|F_t(x_t)\|)\sqrt{\frac{\log(2/\delta)}{K}} = c(x_t)\sqrt{\frac{\log(2/\delta)}{K}}, \tag{94}$$

where we assume sufficiently large $K$ such that $\hat{B} \ge \frac{1}{2}B$.

Now, for sufficiently large $K$, the Taylor expansion of $\hat{F}_t$ is expressed as:

$$\hat{F}_t = \frac{A}{B} + \frac{1}{B}(\hat{A} - A) - \frac{A}{B^2}(\hat{B} - B) - \frac{1}{B^2}(\hat{A} - A)(\hat{B} - B) + \frac{A}{B^3}(\hat{B} - B)^2 + O\left(\frac{1}{K^2}\right).$$

To derive the final equation for the bias term, one can express the bias term as follows:

$$\text{Bias}[\hat{F}_t] = \mathbb{E}[\hat{F}_t] - \frac{A}{B} = -\frac{1}{B^2}\text{Cov}[\hat{A}, \hat{B}] + \frac{A}{B^3}\text{Var}[\hat{B}].$$

Since $\text{Cov}[\hat{A}, \hat{B}] = \text{Cov}\left[\tilde{p}_1 F_{t|1}, \tilde{p}_1\right]/K$ and $\text{Var}[\hat{B}] = \text{Var}[\tilde{p}_1]/K$, one obtains the conclusion for $\text{Bias}[\hat{F}_t]$.

To derive the final equation for the variance term, one can combine the sub-Gaussianity of $\hat{F}_t$ with the error bound on $\left\|\hat{F}_t(x_t) - F_t(x_t)\right\|$. $\qquad\square$

**Bias and variance of bootstrapping estimation.** Now, we derive analogous asymptotic bounds for the bias and variance of the bootstrapping estimator. We assume a fully trained intermediate energy model, which allows us to isolate the impact of estimation bias on the intermediate energy while disregarding any neural network learning bias.

**Proposition 2.** *Consider a fully trained surrogate energy model* $\mathcal{E}_r^\phi(x_r) = \mathbb{E}[\hat{\mathcal{E}}_r(x_r)]$, *where* $\hat{\mathcal{E}}_r(x_r)$ *is a biased energy estimator from Equation (20). Let* $\hat{F}_t(x_t; \mathcal{E}_r)$ *be the bootstrapping estimator using* $\mathcal{E}_r$ *as the intermediate energy function:*

$$\hat{F}_t(x_t; \mathcal{E}_r) = \frac{\sum_i \exp(-\mathcal{E}_r(x_r^{(i)}))F_{t|r}^{x_r^{(i)}}(x_t)}{\sum_i \exp(-\mathcal{E}_r(x_r^{(i)}))}. \tag{95}$$

Then, the bias and the variance bound of the bootstrapping estimator with surrogate model $\hat{F}_t(x_t; \mathcal{E}_r^\phi)$ are given by:

$$Bias[\hat{F}_t(x_t; \mathcal{E}_r^\phi)] = Bias[\hat{F}_t(x_t; \mathcal{E}_r)] + O\left(\frac{1}{K^2}\right),$$

$$Var[\hat{F}_t(x_t; \mathcal{E}_r^\phi)] = Var[\hat{F}_t(x_t; \mathcal{E}_r)] = \frac{Var[\tilde{p}_r(x_r)]}{Var[\tilde{p}_1(x_1)]} Var[\hat{F}_t(x_t)].$$

where $\hat{F}_t(x_t)$ is the estimator without bootstrapping. Since $Var[\tilde{p}_r(x_r)] < Var[\tilde{p}_1(x_1)]$, the variance of bootstrapping estimator is smaller than the $\hat{F}_t(x_t)$.

*Proof.* The bias of the energy estimator $\hat{\mathcal{E}}_r$ is given as follows (refer to Corollary 3.2 of [30]):

$$Bias[\hat{\mathcal{E}}_r(x_r)] = \frac{Var[\tilde{p}_1(x_1)]}{2p_r^2(x_r)K} + O\left(\frac{1}{K^2}\right) \tag{96}$$

Thus, we can approximate the surrogate energy as,

$$\mathcal{E}_r^\phi(x_r) = \mathbb{E}[\hat{\mathcal{E}}_r(x_r)] = \mathcal{E}_r(x_r) + \underbrace{\frac{Var[\tilde{p}_1(x_1)]}{2p_r^2(x_r)K}}_{:=b(x_r)}. \tag{97}$$

Plugging the above equation into the $\hat{F}_t(x_t; \mathcal{E}_r^\phi)$, we get,

$$\hat{F}_t(x_t; \mathcal{E}_r^\phi) = \frac{\sum_i \exp(-\mathcal{E}_r^\phi(x_r^{(i)}))F_{t|r}^{x_r^{(i)}}(x)}{\sum_i \exp(-\mathcal{E}_r^\phi(x_r^{(i)}))} = \frac{\sum_i \exp(-\mathcal{E}_r(x_r^{(i)}) - b(x_r^{(i)}))F_{t|r}^{x_r^{(i)}}(x)}{\sum_i \exp(-\mathcal{E}_r(x_r^{(i)}) - b(x_r^{(i)}))}. \tag{98}$$

Here, $b(x_r^{(i)})$ is close to 0 and concentrated to:

$$m_b = b(x_t) = \frac{Var[\tilde{p}_1(x_1)]}{2p_r^2(x_t)K}, \tag{99}$$

when $t$ is close to $r$ and $K$ is sufficiently large. To keep notation concise, let $w_i$ be the self-normalized importance weight with true energy $\mathcal{E}_r(x_r)$ and $F_{t|r}^{(i)}$ be conditional generator with the sample $x_r^{(i)}$:

$$w_i = \frac{\exp(-\mathcal{E}_r(x_r^{(i)}))}{\sum_j \exp(-\mathcal{E}_r(x_r^{(j)}))}, \quad F_{t|r}^{(i)} = F_{t|r}^{x_r^{(i)}}(x). \tag{100}$$

Then, by applying first-order Taylor expansion to the $\hat{F}_t(x_t; \mathcal{E}_r^\phi)$ and approximation $b(x_r^{(i)}) \approx m_b$, we obtain:

$$\hat{F}_t(x_t; \mathcal{E}_r^\phi) = \frac{\sum_i w_i \exp(-b(x_r^{(i)}))F_{t|r}^{(i)}}{\sum_i w_i \exp(-b(x_r^{(i)}))} \tag{101}$$

$$\approx \frac{\sum_i w_i(1 - b(x_r^{(i)}))F_{t|r}^{(i)}}{\sum_i w_i(1 - b(x_r^{(i)}))} \tag{102}$$

$$= \frac{\sum_i w_i F_{t|r}^{(i)} - \sum_i w_i b(x_r^{(i)})F_{t|r}^{(i)}}{1 - \sum_i w_i b(x_r^{(i)})} \tag{103}$$

$$\approx \left(\sum_i w_i F_{t|r}^{(i)} - \sum_i w_i b(x_r^{(i)})F_{t|r}^{(i)}\right)\left(1 + \sum_i w_i b(x_r^{(i)})\right) \tag{104}$$

$$\approx \left(\sum_i w_i F_{t|r}^{(i)} - m_b \sum_i w_i F_{t|r}^{(i)}\right)(1 + m_b) \tag{105}$$

$$\approx (1 - m_b^2)\frac{\sum_i \exp(-\mathcal{E}_r(x_r^{(i)}))F_{t|r}^{x_r^{(i)}}(x)}{\sum_i \exp(-\mathcal{E}_r(x_r^{(i)}))} \tag{106}$$

$$= (1 - m_b^2)\hat{F}_t(x_t; \mathcal{E}_r) \tag{107}$$

Therefore, the bias of the bootstrapping estimator $\hat{F}_t(x_t; \mathcal{E}_r^\phi)$ is:

$$\text{Bias}[\hat{F}_t(x_t; \mathcal{E}_r^\phi)] = \mathbb{E}[\hat{F}_t(x_t; \mathcal{E}_r^\phi)] - F_t(x_t) \tag{108}$$

$$= (1 - m_b^2)\mathbb{E}[\hat{F}_t(x_t; \mathcal{E}_r)] - F_t(x_t) \tag{109}$$

$$= (1 - m_b^2)(F_t + \text{Bias}[\hat{F}_t(x_t; \mathcal{E}_r)]) - F_t(x_t) \tag{110}$$

$$= (1 - m_b^2)\text{Bias}[\hat{F}_t(x_t; \mathcal{E}_r)] - m_b^2 F_t(x_t) \tag{111}$$

$$= \text{Bias}[\hat{F}_t(x_t; \mathcal{E}_r)] - \frac{v_{1r}^2(x_t)}{4p_r^4(x_t)K^2}\left(F_t(x_t) + \text{Bias}[\hat{F}_t(x_t; \mathcal{E}_r)]\right) \tag{112}$$

$$= \text{Bias}[\hat{F}_t(x_t; \mathcal{E}_r)] + O\left(\frac{1}{K^2}\right) \tag{113}$$

Similarly, the variance of the bootstrapping estimator is:

$$\text{Var}[\hat{F}_t(x_t; \mathcal{E}_r^\phi)] = (1 - m_b^2)^2 \text{Var}[\hat{F}_t(x_t; \mathcal{E}_r)] \approx \text{Var}[\hat{F}_t(x_t; \mathcal{E}_r)] \tag{114}$$

since $m_b < 1$ for sufficiently large $K$.

The variance of the IS estimator $\hat{F}_t(x_t)$ without bootstrapping is given by:

$$\text{Var}[\hat{F}_t(x_t)] = \frac{4\text{Var}[\tilde{p}_1(x_r)]}{p_t^2(x_t)K}(1 + \|F_t(x_t)\|)^2, \tag{115}$$

Also, the variance of the bootstrapping estimator $\hat{F}_t(x_t; \mathcal{E}_r)$ with true energy is given by:

$$\text{Var}[\hat{F}_t(x_t; \mathcal{E}_r)] = \frac{4\text{Var}[\tilde{p}_r(x_r)]}{p_t^2(x_t)K}(1 + \|F_t(x_t)\|)^2, \tag{116}$$

Consequently,

$$\text{Var}[\hat{F}_t(x_t; \mathcal{E}_r^\phi)] \approx \text{Var}[\hat{F}_t(x_t; \mathcal{E}_r)] = \frac{\text{Var}[\tilde{p}_r(x_r)]}{\text{Var}[\tilde{p}_1(x_1)]}\text{Var}[\hat{F}_t(x_t)]. \tag{117}$$

Because $\text{Var}[\tilde{p}_r(x_r)] \ll \text{Var}[\tilde{p}_1(x_1)]$, the variance of the bootstrapping estimator is smaller than the $\hat{F}_t(x_t)$. Consequently, with an fully trained energy model, we reduce the variance of the SNIS generator estimation. $\qquad\square$

# B Generator estimation in the multimodal spaces

Our estimator can also be applied to the mixed state spaces $S = X \times Y$ within the generator matching framework. Let $\{\tilde{p}_{t|1}(\cdot|x_1)\}_{0 \leq t \leq 1}$ and $\{\bar{p}_{t|1}(\cdot|y_1)\}_{0 \leq t \leq 1}$ denote the conditional probability paths on the $X$ and $Y$, respectively, and let $\tilde{\mathcal{L}}_{t|1}^{x_1}$ and $\bar{\mathcal{L}}_{t|1}^{y_1}$ denote the corresponding conditional generators for $x_1 \in X$ and $y_1 \in Y$. Assume these generators are parameterized by $\tilde{F}_{t|1}^{x_1} : [0,1] \times S \to V_1$ and $\bar{F}_{t|1}^{y_1} : [0,1] \times S \to V_2$, respectively. For the joint space $S = X \times Y$, we consider the factorized conditional path:

$$p_{t|1}(dx_t, dy_t|x_1, y_1) := \tilde{p}_{t|1}(dx_t|x_1)\, \bar{p}_{t|1}(dy_t|y_1),$$

where $x_t, x_1 \in X$ and $y_t, y_1 \in Y$.

According to Proposition 5 in Holderrieth et al. [19], the conditional generator associated with $p_{t|1}$ admits the following parameterization:

$$F_{t|1}^{x_1, y_1}(x_t, y_t) = \left( \tilde{F}_{t|1}^{x_1}(x_t), \bar{F}_{t|1}^{y_1}(y_t) \right),$$

where the sum, scalar product, and inner product are naturally defined over the tuple $(\cdot, \cdot) \in V_1 \times V_2$. Thus, the importance sampling estimator for the parameterized generator can be written as:

$$F_t(x_t, y_t) = \mathbb{E}_{x_1, y_1 \sim p_{1|t}(\cdot|x_t, y_t)} \left[ F_{t|1}^{x_1, y_1}(x_t, y_t) \right], \tag{118}$$

$$= \mathbb{E}_{x_1, y_1 \sim q_{1|t}(\cdot|x_t, y_t)} \left[ \frac{dp_{1|t}}{dq_{1|t}}(x_1, y_1|x_t, y_t) \left( \tilde{F}_{t|1}^{x_1}(x_t), \bar{F}_{t|1}^{y_1}(y_t) \right) \right]. \tag{119}$$

As in the uni-modality case, this leads to a self-normalized importance sampling estimator, which can be directly extended to the bootstrapping setting. This demonstrates the generality and flexibility of our framework in handling multi-modal spaces.

# C Example of EGM with application to flow and masked diffusion

## C.1 Generator of flow and jump model

In this section, we provide the definition of flow and discrete jump models, their generators and parametrizations. For the case of diffusion processes or more rigorous derivations, we refer the reader to Holderrieth et al. [19]. The discrete jump model is often referred to as a continuous-time Markov chain (CTMC).

**Flow model.** Let the state space be $S = \mathbb{R}^d$, and let $u_t : \mathbb{R}^d \times [0,1] \to \mathbb{R}^d$ be a time-dependent vector field. The flow $X_t$ is defined by the following ordinary differential equation:

$$\frac{dX_t}{dt} = u_t(X_t), \quad X_0 \sim p_0. \tag{120}$$

By definition of the generator, the generator of the flow model is given by

$$\mathcal{L}_t f(x) = \lim_{h \to 0} \frac{\mathbb{E}[f(X_{t+h})|X_t = x] - f(x)}{h} \tag{121}$$

$$= \lim_{h \to 0} \frac{\mathbb{E}[f(X_t + hu_t(X_t) + o(h))|X_t = x] - f(x)}{h} \tag{122}$$

$$= \lim_{h \to 0} \frac{\mathbb{E}[f(X_t) + h\nabla f(x)^T u_t(X_t) + o(h)|X_t = x] - f(x)}{h} \tag{123}$$

$$= \nabla f(x)^T u_t(X_t), \tag{124}$$

where we use a first-order Taylor expansion. Hence, the generator of the flow model admits the following linear parametrization:

$$\mathcal{L}_t f(x) = \langle \mathcal{K}f(x), u_t(x) \rangle, \quad \mathcal{K}f(x) = \nabla f(x), \tag{125}$$

i.e., the generator is parameterized by the ODE vector field $u_t$, and EGM aims to learn $u_t$ via its conditional counterpart $u_{t|1}$.

**Discrete jump model.** Let the state space $S$ be discrete with $|S| < \infty$, and define the time-dependent transition rate matrix $Q_t : S \times S \times [0,1] \to \mathbb{R}$ such that $Q_t(x,x) = -\sum_{y \neq x} Q_t(y,x)$ and $Q_t(y,x) \geq 0$ for all $y \neq x$. The CTMC is defined by the transition rule:

$$X_{t+h} \sim p_{t+h|t}(\cdot|X_t) = \delta_{X_t}(\cdot) + hQ_t(\cdot, X_t). \tag{126}$$

We derive the generator informally; see Davis [10] for a formal treatment:

$$\mathcal{L}_t f(x) = \lim_{h \to 0} \frac{\mathbb{E}[f(X_{t+h})|X_t = x] - f(x)}{h} \tag{127}$$

$$= \lim_{h \to 0} \frac{\mathbb{E}[f(X_{t+h}) - f(X_t)|X_t = x, \text{Jump in } [t, t+h)]\mathbb{P}(\text{Jump in } [t, t+h))}{h} \tag{128}$$

$$+ \lim_{h \to 0} \underbrace{\frac{\mathbb{E}[f(X_{t+h}) - f(X_t)|X_t = x, \text{No jump in } [t, t+h)]\mathbb{P}(\text{No jump in } [t, t+h))}{h}}_{=0} \tag{129}$$

$$= \lim_{h \to 0} \frac{\sum_{y \neq x}(f(y) - f(x))(\frac{Q_t(y,x)h}{-Q_t(x,x)h})(-Q_t(x,x)h)}{h} \tag{130}$$

$$= \sum_{y \neq x}(f(y) - f(x))Q_t(y,x) = \sum_{y \in S} f(y)Q_t(y,x) \tag{131}$$

Therefore, the generator of the CTMC can be linearly parameterized as:

$$\mathcal{L}_t f(x) = \langle \mathcal{K}f(x), Q_t(\cdot, x) \rangle, \quad \mathcal{K}f(x) = (f(y) - f(x))_{y \in S}, \quad \langle a, b \rangle := \sum_{y \in S} a_y b_y, \tag{132}$$

i.e., the generator is parameterized by the transition rate matrix $Q_t(\cdot, x)$, and EGM aims to learn $Q_t$ via its conditional form $Q_{t|1}$.

**Remark on linear parametrization.** Under mild regularity conditions (e.g., Feller processes), Holderrieth et al. [19] shows that Markov processes on both discrete and continuous state spaces can be universally expressed via linear parameterizations:

1. **Discrete state space** ($|S| < \infty$): The generator is parameterized by the transition rate matrix $Q_t$, corresponding to a CTMC.
2. **Euclidean space** ($S = \mathbb{R}^d$): The generator is parameterized as a combination of flow, diffusion, and jump components.

This implies that, like GM, EGM is capable of modeling a wide range of Markov processes on both discrete and Euclidean spaces.

## C.2 Application to the conditional OT flow model

This section details the application of the EGM framework to flow models defined via the conditional optimal transport (CondOT) path.

**Definition of the CondOT path.** The conditional OT probability path is defined as:

$$X_t = tX_1 + (1 - t)X_0, \tag{133}$$

where $X_1 \sim p_1$, $X_0 \sim p_0 = \mathcal{N}(0, I)$, and $X_0, X_1$ are independent. It linearly interpolates between a Gaussian prior and the target distribution. By construction, the conditional distribution is given by:

$$p_{t|1}(x_t|x_1) = \mathcal{N}(x_t; tx_1, (1 - t)^2 I). \tag{134}$$

**EGM on the CondOT path.** First, consider a naive implementation of EGM with a simple proposal distribution defined as:

$$q_{1|t}(x_1|x_t) \propto p_{t|1}(x_t|x_1) = \mathcal{N}(x_t; tx_1, (1 - t)^2 I) \tag{135}$$

$$\propto \exp\left(-\frac{\|x_t - tx_1\|_2^2}{2(1 - t)^2}\right) \tag{136}$$

$$= \exp\left(-\frac{\|x_1 - \frac{x_t}{t}\|_2^2}{2\frac{(1-t)^2}{t^2}}\right), \tag{137}$$

which implies that

$$q_{1|t}(x_1|x_t) = \mathcal{N}\left(x_1; \frac{x_t}{t}, \frac{(1 - t)^2}{t^2} I\right). \tag{138}$$

This choice yields a simple importance weight of the form $\tilde{w}(x_t, x_1) = \tilde{p}_1(x_1)/Z_{1|t}(x_t)$.

Using the identity from Equation (51), the estimated vector field $u_t(x_t)$ becomes:

$$u_t(x_t) = \frac{\sum_i \frac{\tilde{p}_1(x_1^{(i)})}{Z_{1|t}(x_t)} u_{t|1}^{x_1^{(i)}}(x_t)}{\sum_i \frac{\tilde{p}_1(x_1^{(i)})}{Z_{1|t}(x_t)}} \tag{139}$$

$$= \frac{\sum_i \tilde{p}_1(x_1^{(i)}) u_{t|1}^{x_1^{(i)}}(x_t)}{\sum_i \tilde{p}_1(x_1^{(i)})}, \tag{140}$$

where $x_1^{(i)} \sim q_{1|t}(\cdot|x_t)$. This is precisely the same estimator used in Woo and Ahn [40] for the flow-based sampler.

**Assumption check for bootstrapping.** Next, we derive the bootstrapping estimator. We construct the backward transition kernel $p_{t|r}$ satisfying the marginal consistency Equation (52):

$$p_{t|r}(x_t|x_r) = \mathcal{N}(x_t; \frac{t}{r} x_r, \sigma_t I), \quad \sigma_t = (1 - t)^2 - \frac{t^2}{r^2}(1 - r)^2. \tag{141}$$

We verify the consistency via:

$$\int p_{t|r}(x_t|x_r) p_{r|1}(x_r|x_1) dx_r = p_{t|1}(x_t|x_1). \tag{142}$$

Using reparameterization tricks $X_t = \frac{t}{r}X_r + \sqrt{\sigma_t}\epsilon_t$, $X_r = rX_1 + (1-r)\epsilon_r$, $\epsilon_t \perp \epsilon_r$, we have:

$$X_t = \frac{t}{r}(rX_1 + (1-r)\epsilon_r) + \sqrt{\sigma_t}\epsilon_t \tag{143}$$

$$= tX_1 + \frac{t}{r}(1-r)\epsilon_r + \sqrt{\sigma_t}\epsilon_t \tag{144}$$

$$\stackrel{d}{=} tX_1 + (1-t)\epsilon_t', \quad \epsilon_t' \sim \mathcal{N}(0, I), \tag{145}$$

where $\stackrel{d}{=}$ denotes that two random variables have same distribution. Thus, marginal consistency holds. The conditional vector field $u_{t|r}$ is defined as:

$$u_{t|r}(x_t|x_r) = \frac{1}{r}x_r + \frac{\dot{\sigma}_t}{2\sigma_t}\left(x_t - \frac{t}{r}x_r\right). \tag{146}$$

This vector field arises naturally from differentiation of the reparameterization:

$$X_t = \frac{t}{r}X_r + \sqrt{\sigma_t}X_0 \implies \dot{X}_t = \frac{1}{r}X_r + \dot{\sqrt{\sigma_t}}X_0 \tag{147}$$

$$= \frac{1}{r}X_r + \frac{\dot{\sqrt{\sigma_t}}}{\sqrt{\sigma_t}}\left(X_t - \frac{t}{r}X_r\right) \tag{148}$$

$$= \frac{1}{r}X_r + \frac{\dot{\sigma}_t}{2\sigma_t}\left(X_t - \frac{t}{r}X_r\right). \tag{149}$$

Now, verify that the conditional vector field $u_{t|r}$ satisfies the marginal consistency Equation (53):

$$\mathbb{E}_{x_r \sim p_{r|1,t}(\cdot|x_1,x_t)}[u_{t|r}(x_t|x_r)] = u_{t|1}(x_t|x_1). \tag{150}$$

With $p_{r|1,t}(x_r|x_1, x_t) = \frac{p_{t|r}(x_t|x_r)p_{r|1}(x_r|x_1)}{p_{t|1}(x_t|x_1)}$ being Gaussian, its mean is explicitly:

$$\mu_{r|1,t}(x_1, x_t) = \frac{t(1-r)^2}{r(1-t)^2}x_t + \frac{r\sigma_t}{(1-t)^2}x_1. \tag{151}$$

Direct calculation confirms consistency:

$$\mathbb{E}_{x_r \sim p_{r|1,t}(\cdot|x_1,x_t)}[u_{t|r}(x_t|x_r)] = \mathbb{E}_{x_r \sim p_{r|1,t}(\cdot|x_1,x_t)}\left[\frac{1}{r}x_r + \frac{\dot{\sigma}_t}{2\sigma_t}\left(x_t - \frac{t}{r}x_r\right)\right] \tag{152}$$

$$= \frac{1}{r}\mathbb{E}_{x_r \sim p_{r|1,t}(\cdot|x_1,x_t)}[x_r] + \frac{\dot{\sigma}_t}{2\sigma_t}\left(x_t - \frac{t}{r}\mathbb{E}_{x_r \sim p_{r|1,t}(\cdot|x_1,x_t)}[x_r]\right) \tag{153}$$

$$= \frac{1}{r}\mu_{r|1,t}(x_1, x_t) + \frac{\dot{\sigma}_t}{2\sigma_t}\left(x_t - \frac{t}{r}\mu_{r|1,t}(x_1, x_t)\right). \tag{154}$$

The first term $\frac{1}{r}\mu_{r|1,t}$ reduces to,

$$\frac{1}{r}\mu_{r|1,t}(x_1, x_t) = \frac{t(1-r)^2}{r^2(1-t)^2}x_t + \frac{\sigma_t}{(1-t)^2}x_1 \tag{155}$$

$$= \frac{t(1-r)^2 x_t + r^2\sigma_t x_1}{r^2(1-t)^2} \tag{156}$$

$$= \frac{t(1-r)^2}{r^2(1-t)^2}(x_t - tx_1) + x_1, \tag{157}$$

where we used $\sigma_t = (1-t)^2 - \frac{t^2}{r^2}(1-r)^2$ in the third equality.

The part of second term $x_t - \frac{t}{r}\mu_{r|1,t}(x_1, x_t)$ reduces to,

$$x_t - \frac{t}{r}\mu_{r|1,t}(x_1, x_t) = x_t - \frac{t^2(1-r)^2}{r^2(1-t)^2}x_t - \frac{t\sigma_t}{(1-t)^2}x_1 \tag{158}$$

$$= \frac{r^2(1-t)^2 - t^2(1-r)^2}{r^2(1-t)^2}x_t - \frac{t\sigma_t}{(1-t)^2}x_1 \tag{159}$$

$$= \frac{\sigma_t}{(1-t)^2}x_t - \frac{t\sigma_t}{(1-t)^2}x_1, \tag{160}$$

where we used $\sigma_t = (1-t)^2 - \frac{t^2}{r^2}(1-r)^2$ in the third equality.

Put it all together, we conclude that,

$$\mathbb{E}_{x_r \sim p_{r|1,t}(\cdot|x_1,x_t)}[u_{t|r}(x_t|x_r)] = \frac{1}{r}\mu_{r|1,t}(x_1,x_t) + \frac{\dot{\sigma}_t}{2\sigma_t}(x_t - \frac{t}{r}\mu_{r|1,t}(x_1,x_t)) \tag{161}$$

$$= \frac{t(1-r)^2}{r^2(1-t)^2}(x_t - tx_1) + x_1 + \frac{\dot{\sigma}_t}{2\sigma_t}\left(\frac{\sigma_t}{(1-t)^2}x_t - \frac{t\sigma_t}{(1-t)^2}x_1\right) \tag{162}$$

$$= \frac{t(1-r)^2}{r^2(1-t)^2}(x_t - tx_1) + x_1 + \frac{\dot{\sigma}_t}{2(1-t)^2}(x_t - tx_1) \tag{163}$$

$$= \left(\frac{t(1-r)^2}{r^2} + \frac{\dot{\sigma}_t}{2}\right)\frac{x_t - tx_1}{(1-t)^2} + x_1 \tag{164}$$

$$= (1-t)\frac{x_t - tx_1}{(1-t)^2} + x_1 \tag{165}$$

$$= \frac{x_1 - x_t}{1-t} \tag{166}$$

$$= u_{t|1}(x_t|x_1), \tag{167}$$

which implies the proposed transition kernel $p_{t|r}(x_t|x_r)$ and conditional vector field $u_{t|r}(x_t|x_r)$ satisfies the assumption of our [Theorem 1](#).

**Bootstrapped estimator for the CondOT.** Lastly, we define the bootstrapping estimator for the CondOT flow model using proposal as follows:

$$q_{r|t}(x_r|x_t) \propto p_{t|r}(x_t|x_r) = \mathcal{N}(x_t; \frac{t}{r}x_r, \sigma_t I) \tag{168}$$

$$\propto \exp\left(-\frac{\|x_t - \frac{t}{r}x_r\|_2^2}{2\sigma_t}\right) \tag{169}$$

$$\propto \exp\left(-\frac{\|x_r - \frac{r}{t}x_t\|_2^2}{2\frac{r^2}{t^2}\sigma_t}\right), \tag{170}$$

which implies that

$$q_{r|t}(x_r|x_t) = \mathcal{N}\left(x_r; \frac{r}{t}x_t, \frac{r^2}{t^2}\sigma_t I\right). \tag{171}$$

The bootstrapping estimator is then given by:

$$\hat{u}_t(x_t) = \frac{\sum_{i=1}^K \tilde{w}(x_t, x_r^{(i)})u_{t|r}(x_t|x_r)}{\sum_{i=1}^K \tilde{w}(x_t, x_r^{(i)})}, \quad \tilde{w}(x_t, x_r) = \tilde{p}_r(x_r) = \exp(-\mathcal{E}_r^\phi(x_r)), \tag{172}$$

where samples $x_r^{(1)}, \ldots, x_r^{(K)} \sim q_{r|t}(\cdot|x_t)$ and $\mathcal{E}_r^\phi(x_r)$ is learned energy estimator.

### C.3 Application to the masked diffusion model

This section describes how the EGM framework can be applied to discrete jump models using the masked diffusion path.

**Definition of masked diffusion path.** We define the masked diffusion path as follows:

$$p_{t|r}(x_t|x_r) = \frac{\kappa_t}{\kappa_r}\delta_{x_r}(x_t) + \left(1 - \frac{\kappa_t}{\kappa_r}\right)\delta_M(x_t). \tag{173}$$

where $\kappa_t : [0,1] \to \mathbb{R}_{>0}$ is an increasing function satisfying $\kappa_0 = 0, \kappa_1 = 1$, $M$ is the mask token, and $\delta_x$ is the Dirac-delta distribution centered at $x$. Next, we derive the conditional transition rate matrix generating the conditional probability path $p_{t|r}(\cdot|x_r)$. Starting from the Kolmogorov forward

equation, we have:

$$\frac{d}{dt}p_{t|r}(y_t|x_r) = \frac{\dot{\kappa}_t}{\kappa_r}(\delta_{x_r}(y_t) - \delta_M(y_t)) \tag{174}$$

$$= \frac{\dot{\kappa}_t}{\kappa_r}\frac{1}{\kappa_r - \kappa_t}((\kappa_r - \kappa_t)\delta_{x_r}(y_t) - (\kappa_r - \kappa_t)\delta_M(y_t)) \tag{175}$$

$$= \frac{\dot{\kappa}_t}{\kappa_r}\frac{\kappa_r}{\kappa_r - \kappa_t}(\delta_{x_r}(y_t) - p_{t|r}(y_t|x_r)) \tag{176}$$

$$= \sum_{x_t}\frac{\dot{\kappa}_t}{\kappa_r - \kappa_t}(\delta_{x_r}(y_t) - \delta_{x_t}(y_t))p_{t|r}(x_t|x_r) \tag{177}$$

$$= \sum_{x_t}u_{t|r}(y_t, x_t|x_r)p_{t|r}(x_t|x_r), \tag{178}$$

thus obtaining $u_{t|r}(y_t, x_t|x_r) = \frac{\dot{\kappa}_t}{\kappa_r - \kappa_t}(\delta_{x_r}(y_t) - \delta_{x_t}(y_t))$.

**EGM on the masked diffusion path.** We first introduce a naive implementation of EGM using a simple proposal distribution defined as:

$$q_{1|t}(x_1|x_t) \propto p_{t|1}(x_t|x_1) = \kappa_t\delta_{x_1}(x_t) + (1 - \kappa_t)\delta_M(x_t) \tag{179}$$

$$\tag{180}$$

which implies:

$$q_{1|t}(x_1|x_t) = \begin{cases} \text{Unif}(x; S - M) & (x = M) \\ \delta_{x_t}(x_1) & (x \neq M) \end{cases}. \tag{181}$$

This yields the simple importance weight $\tilde{w}(x_t, x_1) = \tilde{p}_1(x_1)/Z_{1|t}(x_t)$. Following Equation (51), the estimator for the transition matrix $u_t(y_t, x_t)$ becomes:

$$\hat{u}_t(y_t, x_t) = \frac{\sum_{i=1}^K \tilde{p}_1(x_1)u_{t|1}(y_t, x_t|x_1^{(i)})}{\sum_{i=1}^K \tilde{p}_1(x_1^{(i)})} \tag{182}$$

where samples $x_1^{(1)}, \ldots, x_1^{(K)} \sim q_{1|t}(\cdot|x_t)$.

In practice, the state space $S = [N]^D$ factorizes along dimensions, where $D$ is sequence length and $[N] = \{1, \ldots, N\}$. We thus factorize the masked diffusion path as follows:

$$p_{t|1}(x_t|x_1) = \prod_{i=1}^D p_{t|1}^i(x_t^i|x_1^i), \quad p_{t|1}^i(x_t^i|x_1^i) = \kappa_t\delta_{x_1^i}(x_t^i) + (1 - \kappa_t)\delta_M(x_t^i) \tag{183}$$

where $x^i \in [N]$ denotes the $i$-th token of the sequence $x \in S$. The proposal $q_{1|t}$ and the transition rate matrix $u_t(y, x)$ factorize accordingly. The proposal factorizes as:

$$q_{1|t}(x_1|x_t) = \prod_{i=1}^D q_{1|t}^i(x_1^i|x_t^i) \propto \prod_{i=1}^D p_{t|1}^i(x_t^i|x_1^i), \tag{184}$$

where $q_{1|t}^i$ is a proposal defined over each dimensions. The transition matrix factorizes as:

$$u_t(y, x) = \sum_{i=1}^D \delta(y^{-i}, x^{-i})u_t^i(y^i, x), \tag{185}$$

where $x^{-i}$ denotes the $x$ without $i$-th token and $u_t^i$ is transition rate for each dimension. Hence, our neural network is trained to predict the $D \times N$ matrix $\text{NN}_\theta : (x_t, t) \mapsto (u_t^i(y_t^i, x_t))_{1 \leq i \leq D, \, y_t^i \in [N]}$.

**Assumption check for bootstrapping.** The backward transition kernel $p_{t|r}$ of masked diffusion satisfies the marginal consistency since it defines the Markov process (noising process of masked diffusion). Thus, it is suffice to show that the conditional transition rate matrix $u_{t|r}(y_t, x_t|x_r)$ satisfies the marginal consistency Equation (53):

$$\mathbb{E}_{x_r \sim p_{r|1,t}(\cdot|x_1, x_t)}[u_{t|r}(y_t, x_t|x_r)] = u_{t|1}(y_t, x_t|x_1). \tag{186}$$

This condition can be confirmed via explicit calculations. Note that $p_{r|1,t}(x_r|x_1, x_t) = \frac{p_{t|r}(x_t|x_r)p_{r|1}(x_r|x_1)}{p_{t|1}(x_t|x_1)}$.

$$(\text{L.H.S.}) = \sum_{x_r} \frac{\dot{\kappa}_t}{\kappa_r - \kappa_t} (\delta_{x_r}(y_t) - \delta_{x_t}(y_t)) p_{r|1,t}(x_r|x_1, x_t) \tag{187}$$

$$= \sum_{x_r} \frac{\dot{\kappa}_t}{\kappa_r - \kappa_t} (\delta_{x_r}(y_t) - \delta_{x_t}(y_t)) \frac{p_{t|r}(x_t|x_r)p_{r|1}(x_r|x_1)}{p_{t|1}(x_t|x_1)} \tag{188}$$

$$= \frac{\dot{\kappa}_t}{(\kappa_r - \kappa_t)p_{t|1}(x_t|x_1)} \sum_{x_r} (\delta_{x_r}(y_t) - \delta_{x_t}(y_t)) p_{t|r}(x_t|x_r)p_{r|1}(x_r|x_1) \tag{189}$$

$$= \frac{\dot{\kappa}_t}{(\kappa_r - \kappa_t)p_{t|1}(x_t|x_1)} \Big( (\delta_M(y_t) - \delta_{x_t}(y_t))p_{t|r}(x_t|M)p_{r|1}(M|x_1) \tag{190}$$

$$+ (\delta_{x_1}(y_t) - \delta_{x_t}(y_t))p_{t|r}(x_t|x_1)p_{r|1}(x_1|x_1) \Big) \tag{191}$$

$$= \frac{\dot{\kappa}_t}{(\kappa_r - \kappa_t)p_{t|1}(x_t|x_1)} \Big( (\delta_M(y_t) - \delta_{x_t}(y_t))\delta_M(x_t)(1 - \kappa_r) \tag{192}$$

$$+ (\delta_{x_1}(y_t) - \delta_{x_t}(y_t))p_{t|r}(x_t|x_1)\kappa_r \Big) \tag{193}$$

$$= \begin{cases} 0 & (x_t = x_1) \\ \frac{\dot{\kappa}_t}{(\kappa_r - \kappa_t)(1 - \kappa_t)} (\delta_{x_1}(y_t) - \delta_{x_t}(y_t)) \left(1 - \frac{\kappa_t}{\kappa_r}\right) \kappa_r & (x_t = M) \end{cases} \tag{194}$$

$$= \frac{\dot{\kappa}_t}{1 - \kappa_t}(\delta_{x_1}(y_t) - \delta_{x_t}(y_t)) \tag{195}$$

$$= u_{t|1}(y_t, x_t|x_1) \tag{196}$$

where we used the fact that $p_{r|1}(x_r|x_1) > 0$ for only $x_r = M$ or $x_r = x_1$ in the fourth equality. Hence, the proposed transition kernel $p_{t|r}$ and conditional transition rate matrix $u_{t|r}$ satisfies the assumption of our Theorem 1.

**Bootstrapped estimator for masked diffusion.** Lastly, we define the bootstrapping estimator for the transition rate matrix of masked diffusion model. We use the following proposal:

$$q_{r|t}(x_r|x_t) \propto p_{t|r}(x_t|x_r) = \frac{\kappa_t}{\kappa_r}\delta_{x_r}(x_t) + \left(1 - \frac{\kappa_t}{\kappa_r}\right)\delta_M(x_t) \tag{197}$$

$$= \begin{cases} \frac{\kappa_t}{\kappa_r} & (x_t \neq M, x_r = x_t) \\ 0 & (x_t \neq M, x_r \neq x_t) \\ 1 - \frac{\kappa_t}{\kappa_r} & (x_t = M, x_r \neq M) \\ 1 & (x_t = M, x_r = M) \end{cases} \tag{198}$$

which implies,

$$q_{r|t}(x_r|x_t) = \begin{cases} \delta_{x_t}(x_r) & (x_t \neq M) \\ \text{Cat}(1 - \frac{\kappa_t}{\kappa_r}, \ldots, 1 - \frac{\kappa_t}{\kappa_r}, 1) & (x_t = M) \end{cases} \tag{199}$$

where the mask token is the last token $M = N$ and Cat is the categorical distribution with unnormalized weight.

The bootstrapping estimator is given by:

$$\hat{u}_t(y_t, x_t) = \frac{\sum_{i=1}^{K} \tilde{w}(x_t, x_r^{(i)})u_{t|r}(y_t, x_t|x_r^{(i)})}{\sum_{i=1}^{K} \tilde{w}(x_t, x_r^{(i)})}, \quad \tilde{w}(x_t, x_r) = \tilde{p}_r(x_r) = \exp(-\mathcal{E}_r^\phi(x_r)), \tag{200}$$

where samples $x_r^{(1)}, \ldots, x_r^{(K)} \sim q_{r|t}(\cdot|x_t)$ and $\mathcal{E}_r^\phi(x_r)$ is learned energy estimator.

**Learning energy with generalized NEM objective.** We train the $\mathcal{E}_r^\phi$ with the following estimator for the intermediate energy:

$$\mathcal{E}_r(x_r) = -\log \mathbb{E}_{x_1 \sim q_{1|r}(\cdot|x_r)}[\exp(-\mathcal{E}_1(x_1))] - \log Z_{1|r}(x_r). \tag{201}$$

For masked diffusion, the partition function $Z_{1|r}(x_r)$ explicitly depends on $x_r$, is given by:

$$Z_{1|r}(x_r) = \sum_{x_1} p_{r|1}(x_r|x_1) \tag{202}$$

$$= \sum_{x_1} \kappa_r \delta_{x_1}(x_r) + (1 - \kappa_r)\delta_M(x_r) \tag{203}$$

$$= \begin{cases} (N-1)(1-\kappa_t) & (x_r = M) \\ \kappa_r & (x_r \neq M) \end{cases} \tag{204}$$

where $N$ is the number of token in the state space $S = [N]^D$.

# D  Additional details on the experiments

In this section, we provide detailed descriptions of the experimental tasks, evaluation metrics, and experimental setups used throughout this work. The code is available at here.

## D.1  Task details

**Discrete Ising model.** We consider the Ising model defined on a 2D grid $\{-1, 1\}^{L \times L}$ with size $L$. The energy function $\mathcal{E} : \{-1, 1\}^{L \times L} \to \mathbb{R}$ is given by:

$$\mathcal{E}(x) = \beta \left( -J \sum_{\langle i,j \rangle} x_i x_j + \mu \sum_i x_i \right), \tag{205}$$

where $\langle i, j \rangle$ denotes pairs of neighboring spins, $J$ is the interaction strength, $\mu$ is the magnetic moment, and $\beta$ is the inverse temperature. We employ periodic boundary conditions and specifically focus on the ferromagnetic setting ($J > 0$) without external fields ($\mu = 0$), reducing the energy function to:

$$\mathcal{E}(x) = -\beta J \sum_{\langle i,j \rangle} x_i x_j. \tag{206}$$

We fix the interaction strength at $J = 1.0$ and examine various temperatures through $\beta$.

For evaluation, approximate ground truth samples are generated using an extended Gibbs sampling run consisting of 10k burn-in steps, thinning every 10 steps, and 4 parallel chains, collecting 300k samples in total.

**GB-RBM.** The Gaussian-Bernoulli Restricted Boltzmann Machine (GB-RBM) task involves two continuous visible units following Gaussian distributions and three binary hidden units following Bernoulli distributions, with the energy function:

$$\mathcal{E}(x_1, x_2) = \Sigma^{-1} \|x_1 - a\|_2^2 - \langle b, x_2 \rangle - \Sigma^{-1} x_1^T W x_2, \tag{207}$$

where $x_1, a \in \mathbb{R}^2$, $x_2, b \in \{0, 1\}^3$, $\Sigma \in \mathbb{R}$, and $W \in \mathbb{R}^{2 \times 3}$. Parameters are selected to induce multiple modes, specifically six modes in continuous dimensions (see Figure 5). We set:

$$a = [0, 0], \quad b = [-5, -5, -5], \quad \Sigma = 2, \quad W = \begin{pmatrix} 10 & 0 & 10 \\ 0 & 10 & 0 \end{pmatrix}. \tag{208}$$

Approximately ground truth samples are generated using Gibbs sampling with 10k burn-in steps, 100-step thinning intervals, and 100 parallel chains, collecting 100k samples.

**JointDW4.** JointDW4 exemplifies the molecular sequence-structure co-generation task, extending the classical four-particle double-well (DW4) benchmark with particle-type-dependent interactions. This setup includes 4 particles in 2D space, each assigned discrete types, yielding a 12-dimensional (8 continuous, 4 discrete) energy function:

$$\mathcal{E}_{\text{JointDW4}}(x, t) = \frac{1}{2\tau} \sum_{i,j} a(t_i, t_j)(d_{ij} - d_0) + b(t_i, t_j)(d_{ij} - d_0)^2 + c(t_i, t_j)(d_{ij} - d_0)^4, \tag{209}$$

where $d_{ij} = \|x_i - x_j\|_2$ is a Euclidean distance between the particle $i, j$ and $t_i \in \{1, 2\}$ is the type of particle $i$. The parameters are set as follows:

$$a(\cdot, \cdot) = 0, \quad \tau = 1, \quad d_0 = 2 \quad, b = \begin{pmatrix} -3.0 & -2.5 \\ -2.5 & -2.8 \end{pmatrix}, \quad c = \begin{pmatrix} 0.8 & 0.4 \\ 0.4 & 0.6 \end{pmatrix}, \tag{210}$$

where $b(t_i, t_j) = b_{t_i t_j}$ and $c(t_i, t_j) = c_{t_i t_j}$.

Ground truth samples are similarly obtained from Gibbs sampling, running 10k burn-in steps, thinning every 50 steps, across 100 parallel chains, collecting 100k samples in total.

**JointMoG.** The JointMoG extends a Gaussian mixture benchmark commonly used for evaluating diffusion samplers. It includes one continuous dimension $x \in \mathbb{R}$ and one binary dimension $b \in \{-1, 1\}$:

$$\mathcal{E}_{\text{2D-JointMoG}}(x, b) = \frac{1}{2\sigma^2} \|x - b\|_2^2, \tag{211}$$

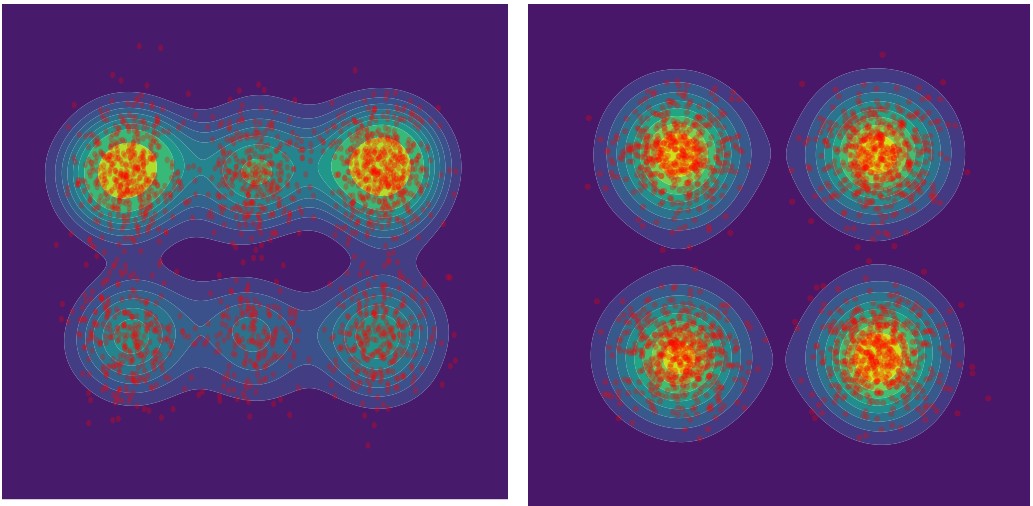

Figure 5: Ground truth sample plot of GB-RBM (left) and JointMoG (right). Samples are projected onto the first two continuous dimensions.

with standard deviation $\sigma$. We scale this model to 20 dimensions (10 continuous, 10 discrete) for benchmarking:

$$\mathcal{E}_{\text{JointMoG}}(x, b) = \sum_i \frac{1}{2\sigma^2} \|x_i - b_i\|_2^2, \tag{212}$$

with $\sigma = 0.3$ to create clearly separated modes. Exact sampling is possible by first sampling discrete variables uniformly and subsequently sampling continuous variables from corresponding Gaussians, providing exact evaluation samples.

### D.2 Metrics

Evaluation metrics in our experiments primarily utilize Wasserstein distances, computed via the Python Optimal Transport (POT) library [13] using exact linear programming. Specifically, we measure the distances between 2000 empirical samples generated by our samplers and 2000 ground truth samples uniformly selected from extensive Gibbs sampling or exact sampling processes.

The Wasserstein distance of order $p$ between two probability measures $\mu$ and $\nu$ is defined as:

$$\mathcal{W}_p(\mu, \nu) = \left( \inf_{\pi \in \prod(\mu, \nu)} \int d(x, y)^p \mathrm{d}\pi(x, y) \right)^{1/p}, \tag{213}$$

where $\prod(\mu, \nu) = \{\pi \in \mathcal{P}(X \times X) \mid \pi(A \times X) = \mu(A), \pi(X \times B) = \nu(B)\}$ is the set of all couplings between $\mu$ and $\nu$, and $d(x, y)$ denotes the metric on the space.

**Energy 1-Wasserstein ($\mathcal{E}$-$\mathcal{W}_1$).** We use the Energy 1-Wasserstein distance as our primary evaluation metric. It measures the 1-Wasserstein distance between the empirical distributions of energy values computed from generated and ground truth samples. This metric is universally applicable across all sampling tasks and effectively captures discrepancies in the energy distributions regardless of the underlying state space and Markov processes involved.

**Magnetization 1-Wasserstein ($M$-$\mathcal{W}1$).** For the discrete Ising model, we additionally employ the magnetization 1-Wasserstein distance. Magnetization for a given spin configuration $x \in \{-1, 1\}^{L \times L}$ is defined as the average spin:

$$M(x) = \frac{1}{L^2} \sum_i x_i. \tag{214}$$

This metric assesses the discrepancy in magnetization distributions between generated and ground truth samples. Particularly in low-temperature scenarios (e.g., $\beta = 0.4$), the system exhibits distinct modes around extreme magnetization values, making this metric especially sensitive to capturing difficulties in multimodal sampling.

Table 4: The best hyper-parameters combination for EGM and Bootstrapping (BS). Flow LR stands for the learning rate for the learned intermediate estimator.

| Tasks | Method | Hidden dim. | # of layers | Flow LR | $\epsilon$ |
|---|---|---|---|---|---|
| Ising $5 \times 5$, $\beta = 0.2$ | EGM | 256 | 3 | - | - |
| | BS | 256 | 3 | $2 \times 10^{-3}$ | 0.05 |
| Ising $5 \times 5$, $\beta = 0.4$ | EGM | 256 | 3 | - | - |
| | BS | 256 | 3 | $10^{-3}$ | 0.05 |
| Ising $10 \times 10$, $\beta = 0.2$ | EGM | 1024 | 3 | - | - |
| | BS | 512 | 3 | $10^{-3}$ | 0.05 |
| Ising $10 \times 10$, $\beta = 0.4$ | EGM | 256 | 3 | - | - |
| | BS | 2048 | 3 | $10^{-3}$ | 0.05 |
| GB-RBM | EGM | 128 | 6 | - | - |
| | BS | 128 | 6 | $10^{-3}$ | 0.01 |
| JointDW4 | EGM | 128 | 6 | - | - |
| | BS | 128 | 6 | $10^{-3}$ | 0.01 |
| JointMoG | EGM | 128 | 6 | - | - |
| | BS | 128 | 6 | $10^{-3}$ | 0.01 |

**Sample 2-Wasserstein** ($x$-$\mathcal{W}_2$). Specifically used for the GB-RBM task, the sample 2-Wasserstein distance evaluates discrepancies between the empirical distributions of generated and ground truth samples projected onto the first two continuous dimensions. A high sample 2-Wasserstein distance coupled with low energy 1-Wasserstein may indicate mode collapse within certain low-energy modes. Due to interpretability concerns (e.g., a poor sampler generating trivial solutions might misleadingly score well), we do not employ this metric for tasks beyond GB-RBM.

### D.3 Experimental setup

We performed a grid search to determine the optimal hyperparameters for each experimental task and method, evaluating each configuration using three random seeds.

As a baseline, we report the performance of a traditional Gibbs sampler [16]. Specifically, we ran Gibbs sampling with four parallel chains, each performing 6000 steps, collecting a total of 24,000 samples. For evaluation purposes, we uniformly subsampled 2000 samples from this set.

Across experiments, we employed 2000 Monte Carlo samples for estimations and a training batch size of 300. Both EGM and bootstrapping utilized 100 outer-loop iterations, with each iteration collecting 2000 samples into a buffer with a maximum size of 10k. The inner-loop iterations were set to 100 for EGM and 1000 for bootstrapping. We adopted a linear masking schedule ($\kappa_t = t$), a linear conditional OT schedule ($\alpha_t = t$), and an exponential variance exploding (VE) schedule ($\sigma_t = \sigma_{\max}(\frac{\sigma_{\min}}{\sigma_{\max}})^t$). All samplers were trained using the AdamW optimizer with an initial learning rate of $10^{-3}$, applying a cosine learning rate schedule with $\eta_{\min} = 10^{-5}$. Training was conducted on an NVIDIA-3090 GPU (24GB VRAM).

For bootstrapping, the intermediate energy model $\mathcal{E}^\phi$ was trained with a separately tuned learning rate. Bootstrapping step sizes of $\epsilon \in \{0.01, 0.05\}$ were evaluated, and an exponential moving average (EMA) was applied to stabilize estimates from $\mathcal{E}^\phi$.

In multi-modal tasks, we applied a weighted loss combining discrete transition rate matrix prediction errors and continuous drift prediction errors: $L_{\text{EGM}} = \lambda_{\text{disc}} L_{\text{discrete}} + \lambda_{\text{conti}} L_{\text{continuous}}$, with fixed coefficients $\lambda_{\text{disc}} = 5.0$ and $\lambda_{\text{conti}} = 1.0$.

Additional task-specific details are provided below, and optimal hyperparameters are summarized in Table 4.

**Discrete Ising model.** We employed a 3-layer MLP with sinusoidal time embeddings for both the intermediate energy function and the transition rate matrix. Each discrete token representing spin values -1 or 1 was embedded in 4 dimensions. Following Gat et al. [14], the transition rate matrix $u_t(y, x)$ was parametrized using a probability denoiser $p_{1|t}(y|x)$ analogous to the $x_1$-prediction in the flow models. Hidden dimensions were explored within $\{256, 512\}$, with additional trials at $\{1024, 2048\}$ for the $10 \times 10$ Ising grid.

**GB-RBM.** We utilized a 6-layer residual MLP with 128 hidden units, a 4-dimensional discrete embedding, and a 64-dimensional continuous embedding. Discrete and continuous embeddings were concatenated and fed into the shared 6-layer MLP. Separate predictor networks subsequently estimated the continuous drift and discrete transition rate matrix. The conditional OT path performed best for both EGM and bootstrapping. We clipped the regression target $F_t$ at a maximum norm of 20 and the energy estimator $\hat{\mathcal{E}}_t$ at 100 to stabilize training.

**JointDW4.** The network architecture matched that used in GB-RBM. The conditional OT path again yielded optimal performance for both methods. Regression targets $F_t$ and energy estimates $\hat{\mathcal{E}}_t$ were clipped at maximum norms of 100 and 1000, respectively.

**JointMoG.** We maintained the same 6-layer residual MLP structure as GB-RBM. The VE path achieved superior performance for both methods, configured with $\sigma_{\max} = 2.0$ and $\sigma_{\min} = 0.01$. The regression targets and energy estimates were clipped to maximum norms of 100 and 1000, respectively.

# E    Limitations and Discussion

We have presented an energy-driven training framework for continuous-time Markov processes (CTMPs). Our method introduces an energy-based importance sampling estimator for the marginal generator and proposes an additional bootstrapping scheme to reduce the variance of importance weights. By lowering this variance, we demonstrate that the bootstrapping approach significantly enhances the sampler's performance.

**Limitations of our work.** Despite the strengths of our approach, several limitations remain. First, we have not extensively evaluated the method on high-dimensional tasks due to limited computational resources. While our framework performs well on benchmarks of moderate scale, its scalability to complex high-dimensional domains—such as protein conformer generation—remains an open question.

Second, we observe that the training process can be unstable. We hypothesize that this instability stems from the simultaneous optimization of the CTMP and the energy model. This joint training often leads to degraded sampling performance. We apply exponential moving average updates to the energy model, which empirically stabilizes training. Nonetheless, further investigation is required to improve robustness.

Third, our estimator incurs bias due to self-normalized importance sampling and the potential mismatch between the proposal and the true posterior distributions. This bias may compromise the accuracy of generator estimation, particularly when the proposal diverges significantly from the posterior. Although the bootstrapping scheme helps reduce this mismatch, its effectiveness depends on the intermediate energy estimator's quality, which may introduce additional bias.

**Comparison to LEAPS.** We compare our method to LEAPS [20], a neural sampler designed for discrete spaces. Our framework is more general in that it applies to arbitrary state spaces and Markov processes, including both continuous and discrete cases, whereas LEAPS is limited to discrete domains. Even when instantiated with a discrete sampler, EGM and LEAPS differ fundamentally. EGM relies on the prescribed conditional probability paths that mix the target distribution, while LEAPS is built on the escorted transport with a temperature annealing. In continuous domains, it has been shown that geometric annealing paths can lead to optimal drifts with high Lipschitz constants [27], which limits sampler performance; whether a similar issue arises in discrete spaces remains an open question.

Additionally, EGM does not utilize an MCMC kernel (analogous to Langevin preconditioning in continuous settings), whereas LEAPS explicitly relies on this mechanism. We believe exploring both directions—leveraging and omitting Langevin preconditioning or MCMC kernels—offers promising avenues for future research.

# F Additional results

## F.1 Additional qualitative results

We provide additional qualitative results for the experiments in Section 4. In Figure 6, we plot the energy histogram of the GB-RBM and JointDW4 compared to the ground truth sample. The Gibbs sampler baseline, EGM, and Bootstrapping match the ground truth energy histogram. However, the Gibbs sampler on GB-RBM suffers from mode collapse as demonstrated in Figure 3.

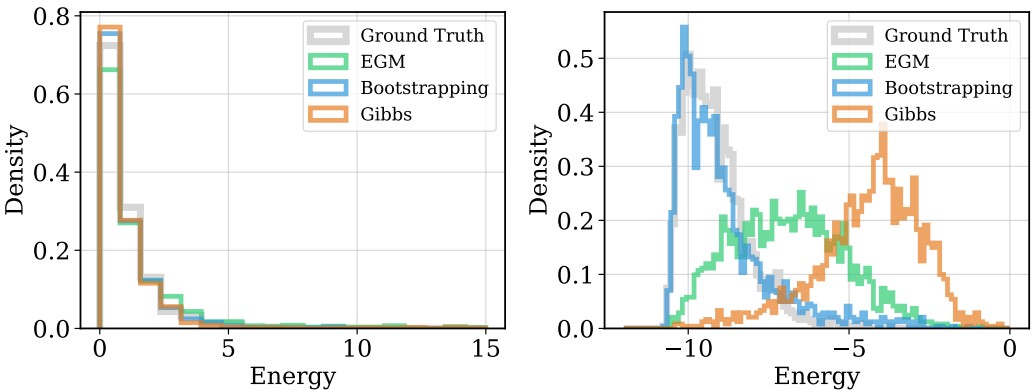

Figure 6: Energy histograms of various samplers on the GB-RBM (left) and JointDW4 (right).

## F.2 Effective sample size of our MC estimator

We present quantitative evidence of the bootstrapped estimator's superiority over the naive EGM. Since the true marginal generator is intractable, we assess estimator quality via the effective sample size (ESS):

$$ESS = \frac{\left(\sum_{i=1}^{n} \tilde{w}_i\right)^2}{\sum_{i=1}^{n} \tilde{w}_i^2} \frac{1}{n} \tag{215}$$

where $\tilde{w}_i$ denotes the unnormalized importance weight with the $i$-th proposed sample and $n$ is the total number of MC samples. We report the *normalized* ESS to indicate the fraction of effectively used samples. Figure 7 shows the average normalized ESS over the course of training. The bootstrapped estimator maintains a significantly higher ESS during training, confirming its improved utilization of proposed samples compared to the naive EGM.

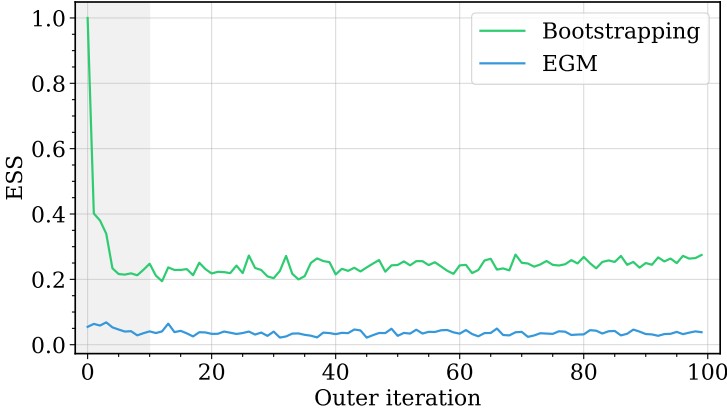

Figure 7: Effective sample size (ESS) of the Monte Carlo estimator during training on the Ising model ($10 \times 10$, $\beta = 0.4$). ESS is evaluated at each regression point and then averaged across all points. In the early training phase (shaded region), the energy model is insufficiently trained, so ESS estimates are unreliable.

### F.3 Additional quantitative results

**Comparison to additional baselines.** Because our proposed method is applicable to *any* state space and *any* Markov process, it can also be directly applied to purely continuous settings, where most neural samplers have been developed. Although many of these existing methods are specialized for continuous domains, we include them as references to contextualize our results. For discrete settings, we additionally include specialized baselines to further demonstrate the performance of our approach. As emphasized earlier, prior neural samplers are typically designed for either continuous or discrete domains and often rely on domain-specific architectures to achieve efficiency, whereas our method imposes no such constraints.

For continuous tasks, we compare against iDEM and PIS on the ManyWell32 benchmark [35]. For discrete tasks, we compare against LEAPS on a $15 \times 15$ Ising model.

Table 5: Performance of *EGM* compared to PIS and iDEM on ManyWell32. BS denotes bootstrapping.

| Algorithm | $\mathcal{E}\text{-}\mathcal{W}_1 \downarrow$ | $x\text{-}\mathcal{W}_2 \downarrow$ | # of modes |
|---|---|---|---|
| **EGM + BS** | 3.30 | 7.07 | 4 |
| **PIS** | 3.39 | 6.14 | 1 |
| **iDEM** | 5.53 | 7.74 | 4 |

Table 6: Performance of *EGM* compared to LEAPS on the Ising model ($15 \times 15$, $\beta = 0.28$). BS denotes bootstrapping.

| Algorithm | $\mathcal{E}\text{-}\mathcal{W}_1 \downarrow$ | $M\text{-}\mathcal{W}_1 \downarrow$ |
|---|---|---|
| **EGM** | 0.89 | 0.06 |
| **EGM + BS** | 0.65 | 0.04 |
| **LEAPS** | 0.68 | 0.01 |
| **LEAPS + MCMC** | 0.49 | 0.01 |

The results show that *EGM* matches or slightly outperforms existing diffusion-based samplers (iDEM, PIS) in continuous settings. In the discrete case, LEAPS performs comparably to *EGM+BS* (bootstrapping). Notably, LEAPS combined with MCMC achieves slightly better performance than *EGM+BS*, albeit at the cost of significantly more energy evaluations (45B vs. 85B).

**Computational complexity analysis.** For reference, Table 7 summarizes the computational complexity of *EGM* compared to a classical Gibbs sampler. We report the number of energy evaluations, wall-clock time per training epoch, and GPU memory usage on a $10 \times 10$ Ising model with $\beta = 0.4$. All experiments are conducted on an NVIDIA RTX 3090 GPU.

Table 7: Computational complexity of *EGM* compared to the Gibbs sampler.

| Method | # Energy evals | Wall-clock time (1 epoch) | Memory footprint |
|---|---|---|---|
| EGM | 200M | 0.16 s | 12 GB |
| EGM + BS | 200M | 0.038 s | 12 GB |
| Parallel Gibbs | 280K | – | 1 GB |

**Effect of bootstrapping time step.** We also study how performance varies with the time gap $\epsilon = r - t > 0$ between two time steps $r$ and $t$. Table 8 reports the results on a $5 \times 5$ Ising model with $\beta = 0.4$.

Table 8: Effect of the bootstrapping time gap $\epsilon$ on performance.

| $\epsilon$ | $\mathcal{E}\text{-}\mathcal{W}_1 \downarrow$ | $M\text{-}\mathcal{W}_1 \downarrow$ |
|---|---|---|
| 0.01 | 2.84 | 0.32 |
| 0.02 | 0.96 | 0.09 |
| 0.05 | 0.84 | 0.08 |
| 0.10 | 0.58 | 0.08 |
| 0.20 | 1.90 | 0.09 |
| 0.50 | 3.73 | 0.20 |

As $\epsilon$ decreases, bootstrapping generally improves performance. However, if $\epsilon$ becomes too small, the scale of the conditional generator $F_{t|r}$ can grow rapidly, causing large fluctuations in loss magnitude

across time steps. This leads to unstable neural network optimization due to inconsistent gradient scales. Consequently, choosing a *moderately small* $\epsilon$ is critical for stable training.

**EMA and training stability.**  Finally, we examine the effect of applying an exponential moving average (EMA) to the parameters of the energy model. EMA is a standard stabilization technique in reinforcement learning, used to mitigate the moving-target problem when regressing on a learned value function. A similar effect is observed here: applying EMA significantly stabilizes training when the generator is conditioned on a learned energy model. Table 9 shows results with and without EMA on a $10 \times 10$ Ising model with $\beta = 0.4$. All other hyperparameters are fixed.

Table 9: Effect of EMA on training stability.

|  | $\mathcal{E}\text{-}\mathcal{W}_1 \downarrow$ | $M\text{-}\mathcal{W}_1 \downarrow$ |
|---|---|---|
| With EMA | $2.51_{\pm 0.16}$ | $0.24_{\pm 0.01}$ |
| Without EMA | $10.08_{\pm 8.65}$ | $0.45_{\pm 0.13}$ |

Using EMA substantially reduces the variance of both energy-based and magnetization metrics, confirming its effectiveness in improving training stability.

