# OpenReview forum: "Energy-based generator matching: A neural sampler for general state space"
_NeurIPS.cc/2025/Conference — NeurIPS 2025 poster_

### Official Review · Reviewer_UQ9j · 2025-06-09

**Clarity:** 4
**Significance:** 3
**Originality:** 3
**Rating:** 4
**Confidence:** 5

**Summary:**

The paper addresses the problem of drawing independent samples from an unnormalised target distribution $p_{\rm target}(x)\propto \exp(-E(x))$ when only an energy function E(x) is available. Building on the recent generator matching (GM) framework, which trains continuous-time Markov processes (CTMPs) to follow a prescribed probability path via minimisation of a Bregman divergence between marginal and conditional generators, the authors introduce and energy-driven training of CTMPs by estimating the marginal generator via self-normalised importance sampling (SNIS).  Under this approach, a proposal distribution $q_{1|t}$ is used to draw candidate endpoints, and weights are computed from the energy function to approximate the intractable posterior over data endpoints. One of the challenges with the naive implementation of this scheme is that the variance in the importance weights can be high. To alleviate this issue, the authors introduce a bootstrapping approach to reduce the high variance of importance weights by introducing an intermediate at time $r>t$ proposal. The paper includes an empirical study on synthetic benchmarks up to 100 dimensions (Ising grids at various temperatures) and joint discrete–continuous tasks (GB-RBM, double-well sequence–structure co-generation, mixture-of-Gaussians). In all cases, the proposed EGM scheme appears to outperform a parallel Gibbs baseline in terms of energy-Wasserstein distance.

**Questions:**

1) Choice of $r$: It would seem that, based on Section 3.2, the method works well when $r$ and $t$ are close, and therefore their densities are similar. However, for the experiments I do not believe $r$ (or rather $\epsilon$) was stated in the main paper. It would be interesting to test the method's efficacy as a function of $\epsilon$.

2) Comparisons to generator matching: I was quite surprised that the paper does not compare against generator matching, flow matching, or any other method (with the exception of Gibbs sampling). Can the authors explain why they haven't compared against other than Gibbs sampling?

3) Real-world examples: Can the authors explain why they haven't used a real-world datasets within their empirical study?

**Ethical Concerns:**

["NO or VERY MINOR ethics concerns only"]

**Final Justification:**

I appreciate the authors' response to my queries and for the extended results. However, as I noted to the authors, I still believe this paper sits on the border of acceptance and I maintain that position.

**Limitations:**

The authors do not really address the limitations of their method. Had they explored more comparisons to alternative approaches, this might have led them to better understand the settings under which this method may not perform as well as competing approaches.

The authors do not discuss the negative societal impacts of their work. However, for this type of paper, I do not believe that this a significant issue.

**Quality:**

4

**Strengths And Weaknesses:**

$\underline{\text{Quality}}$

The paper is methodologically and theoretically sound. The authors present very clear derivations of all of the important concepts in the paper, e.g. importance sampling, generator matching, etc. without any obvious omissions. The paper is very well-structure and easy to read. The method itself is clearly described in the algorithm box, making it easy to recreate the methodology. The empirical study is convincing, in terms of how the method compares to the Gibbs sampler approach.

However, there are a few weaknesses that might be possible to address. The numerics are a bit simple. The paper omits any systematic analysis of computational complexity. There is no clear guidance on the number of energy evaluations required, the wall-clock time per training epoch, or the memory footprint relative to baselines like Gibbs sampling or diffusion-based methods. There is also a lack of comparisons to other methods, e.g. generator matching (with the exception of Figure 1). Additionally, while variance reduction via self-normalised importance sampling does seem to offer an improvement, the bias introduced by this scheme, and also the surrogate energy approximation, is only discussed informally. Currently, there are no theoretical bias bounds, nor does the empirical study show how bias scales with the number of importance samples, or how surrogate accuracy impacts the results. Finally, all empirical validation is confined to synthetic tasks in at most 100 dimensions, leaving the method’s performance on large-scale, high-dimensional, or real-world problems unknown.

$\underline{\text{Clarity}}$

Throughout the manuscript, notation is clear and terms such as: variables, generators, and kernels are introduced in a systematic fashion. The notation also follows a similar style to other papers in this area, which helps readers to compare this paper to other similar works. The figures, especially Figure 1, are well-presented and help to understand the methodology and its relationship with other approaches. The pseudocode in Algorithm 1 offers a clear, step-by-step recipe that helps with comprehension and implementation.

Overall, the paper is very clear, and I think readers who have read the related papers, e.g. flow matching, will find it easy to follow this paper. However, some of the notation occasionally is a bit complicated (but probably hard to avoid) e.g.  symbols such as $\tilde p$, w, and multiple kernels appear in rapid succession, forcing readers to frequently backtrack to distinguish unnormalised densities from importance weights. Furthermore, some critical results, especially those concerning the intractable backward kernels and the precise role of the surrogate energy, are relegated almost entirely to the appendices.

$\underline{\text{Significance}}$

A key contribution of this work is its unified treatment of diverse data modalities. By casting EGM in a measure-theoretic framework, the authors demonstrate how continuous, discrete, and hybrid sampling problems can all be addressed under a single continuous-time Markov process paradigm. This is a nice contribution which addresses a gap in energy-based generative modelling. The bootstrapping trick, which introduces a learned surrogate energy at an intermediate time for variance reduction, represents an interesting approach that has potential utility beyond the specific context of EGM, suggesting broad applicability to other importance-sampling and sequential Monte Carlo methods. Moreover, the method directly addresses scenarios in which only an unnormalised energy is available and no pre-sampled data exist, offering practical value in scientific and engineering settings where normalisation constants cannot always be computed.

However, the true significance remains to be demonstrated in large-scale, applied domains. The current empirical evidence is restricted to synthetic benchmarks, leaving open the question of whether the theoretical benefits translate to real-world gains in fields such as molecule generation or image generation. The empirical comparisons are also narrowly focused on parallel Gibbs sampling; without head-to-head evaluations against cutting-edge alternatives. For example, it's a bit surprising that there are no comparisons with any diffusion-based schemes or flow matching, or even generator matching (especially given that this paper builds on generator matching). Therefore, the claim of broad applicability is weakened. Finally, while the unification of modalities is appealing, some may view the contribution as an incremental extension of existing generator-matching, rather than a unique and novel contribution.

$\underline{\text{Originality}}$

Embedding importance sampling directly into generator matching is a new idea and an interesting methodological contribution. Although generator matching and diffusion-based samplers have been explored previously, to the best of my knowledge, this paper is the first to integrate self-normalised importance sampling in the marginal-generator estimation step, supported by a clear theoretical justification. The introduction of a bootstrapping mechanism via a learned surrogate energy at an intermediate time is quite creative, offering a new approach to control the variance, which I haven't seen before in the energy-based sampling literature.

That said, the work does draw heavily on existing frameworks. The mathematical foundation closely parallels that of iDEM and other diffusion samplers, and the bootstrapping idea, while inventive, relies on surrogate modelling techniques. This is potentially an issue because this now adds an additional layer of approximation which wouldn't be needed if the importance proposal used the simpler $q \propto p$ form. The discussion of connections to GFlowNets and discrete flow models, though present, is somewhat limited (e.g. no numerical comparisons). It would have been interesting to see a deeper exploration of these relationships which could have further highlighted the paper’s distinctive contributions. Ultimately, the originality hinges on the effective combination of the importance sampling, with bootstrapping, within generator matching. This is interesting in itself, but could be seen by some people as a specialised rather than a broadly transformative advance.

---

> ### Author Rebuttal · Authors · 2025-07-31
>
> ### **W1-1: No analysis on computational complexity.**
> To answer your question, we present a table of computational complexity, which includes the number of energy evaluations required, the wall-clock time per training epoch, and the memory footprint on the discrete Ising model 10 x 10, $\beta =0.4$.
>
> | Method   | # of energy evals | Wall-clock time (1 epoch) | Memory footprint |
> | -------- | --------   | -------- | -------- |
> | EGM      | 200M       | 0.16     | 12GB     |
> | EGM + BS | 200M       | 0.038    | 12GB     |
> | Parallel Gibbs | 280K | -        | 1GB      |
>
> ---
> ### **W1-2: The bias and the variance of self-normalized importance sampling are not formally discussed.**
>
> We provide a formal analysis addressing both the bias from self-normalized importance sampling (SNIS) and the surrogate energy approximation.
>
> Specifically, we show that (1) the bias and variance of the SNIS estimator decay as $O(1/K)$, and (2) with a well-trained surrogate energy, the bootstrapping estimator closely matches the bias and variance of the bootstrapping estimator using the true energy. Moreover, bootstrapping reduces the variance compared to the non-bootstrapped estimator.
>
> We state the propositions with proof sketches; for the second proof, see our response to Reviewer meoi (W1). A full rigorous proof will be included in the revised version.
>
> **Proposition.** Consider an unnormalized density $\tilde p_1(x_{1})$ and a conditional generator $F_{t|1}^{x_1}(x_{t})$ evaluated on a sample $x_1 \sim q_{1|t}$. Suppose $\tilde p_1(x_1)$ and $\Big\lVert \tilde p_1(x_1) F_{t|1}^{x_1}(x_t) \Big\rVert$ are sub-Gaussian. Then there exists a constant $c(x_t)$ such that with probability at least $1-\delta$,
> $$
> \left\lVert \hat F_t(x_t) - F_t(x_t) \right\rVert \leq c(x_t) \sqrt{\frac{\log (2/\delta)}{K}},
> $$
> where $\hat F_t(x_t)$ denotes the SNIS estimator from eq. 9 using $K$ samples $x_1^{(1)}, \\ldots, x_1^{(K)}\sim q_{1|t}$:
> $$
> \hat F_t(x_t) = \frac{\sum_{i=1}^K \tilde p_1(x_1^{(i)})F_{t|1}^{x_1^{(i)}}(x_t)}{\sum_{i=1}^K \tilde p_1(x_1^{(i)})}.
> $$
> Moreover, its bias and variance are expressed as:
> $$
> \text{Bias}[\hat F_t] = \frac{1}{Kp_t(x_t)^2} \left(- \text{Cov}\left[\tilde p_1 F_{t|1}, \tilde p_1\right] + F_t(x_t)\text{Var}[\tilde p_1] \right) + O\left(\frac{1}{K^2}\right),
> $$
> $$
> \text{Var}[\hat F_t] = \frac{4\text{Var}[\tilde p_1(x_1)]}{p_t^2(x_t)K}(1 + \lVert F_t(x_t)\rVert)^2,
> $$
> **proof sketch)** By Hoeffding's inequality for sub-Gaussian random variables, there exists a constant $C$ such that
> $$
> \left| \frac{1}{K} \sum_{i=1}^{K} \tilde p_1(x_1^{(i)}) - \tilde p_t(x_t) \right| \leq C \sqrt{\frac{\log (2/\delta) }{K}},
> $$
> $$
> \left\lVert \frac{1}{K} \sum_{i=1}^{K} \tilde p_1(x_1^{(i)}) F_{t|1}^{x_1^{(i)}}(x_t) - \tilde p_t(x_t)F_t(x_t) \right\rVert \leq C \sqrt{\frac{\log (2/\delta) }{K}},
> $$
> with $1 - \delta$ probability. (Here, $C = \sqrt{\text{Var}[\tilde p_1(x_1)]}$ is a possible choice.)
>
> Since we have bounds for both numerator and denominator of $\hat F_t$, we can also bound the error of $\hat F_t$ to $F_t$. The result is as follows:
> $$
> \left\lVert \hat F_t(x_t) - F_t(x_t) \right\rVert \leq \frac{2C}{p_t(x_t)}(1 + \lVert F_t(x_t)\rVert)\sqrt{\frac{\log(2/\delta)}{K}}=c(x_t)\sqrt{\frac{\log(2/\delta)}{K}}.
> $$
> Now, let $\hat{A}$ and $\hat{B}$ denote the numerator and the denominator of $\hat F_t$, respectively. We also let $\mathbb{E}[\hat{A}] = A$ and  $\mathbb{E}[\hat{B}] = B$. For sufficiently large $K$, the Taylor expansion is expressed as:
> $$
> \hat F_t =\frac{A}{B}+\frac{1}{B}(\hat{A}-A) - \frac{A}{B^2}(\hat{B}-B)-\frac{1}{B^2}(\hat{A} - A)(\hat{B} - B) + \frac{A}{B^3} (\hat{B} - B)^2 + O\left(\frac{1}{K^2}\right).
> $$
> To derive the final equation for the bias term, one can express the bias term as follows:
> $$
> \text{Bias}[\hat F_t] = \mathbb{E}[\hat F_t] - \frac{A}{B}=-\frac{1}{B^2}\text{Cov}[\hat{A}, \hat{B}]+ \frac{A}{B^3}\text{Var}[\hat{B}].
> $$
> Since $\text{Cov}[\hat{A}, \hat{B}] = \text{Cov}\left[\tilde p_1 F_{t|1}, \tilde p_1\right] / K$ and $\text{Var}[\hat{B}]=\text{Var}[\tilde p_1]/K$, one obtains the conclusion for $\text{Bias}[\hat F_t]$.
>
> To derive the final equation for the variance term, one can combine the sub-Gaussianity of $\hat F_t$ with the error bound on $\left\lVert \hat F_t(x_t) - F_t(x_t) \right\rVert$.
>
> **Proposition.** Consider an fully trained surrogate energy model $\mathcal{E}_r^{\phi}(x_r) = \mathbb{E}[\hat{\mathcal{E}}_r(x_r)]$, where $\hat{\mathcal{E}}_r(x_r)$ is a biased energy estimator (from eq. 20). Let $\hat F_t(x_t; \mathcal{E}_r)$ be the bootstrapping estimator using $\mathcal{E}_r$ as the intermediate energy function:
> $$
> \hat F_t(x_t;\mathcal E_r) = \frac{\sum\_i \exp (-\mathcal{E}\_r(x_r^{(i)})) F\_{t|r}\^{x_r^{(i)}}(x)}{\sum\_i \exp (-\mathcal{E}\_r(x_r^{(i)}))}.
> $$
> Then, the bias and the variance bound of the bootstrapping estimator with surrogate model $\hat F_t(x_t; \mathcal{E}_r^\phi)$ are given by:
> $$
> \text{Bias}[\hat F_t(x_t;\mathcal{E}_r^\phi)] = \text{Bias}[\hat F_t(x_t;\mathcal{E}_r)] +  O\left(\frac{1}{K^2}\right),
> $$
> $$
> \text{Var}[\hat F_t(x_t;\mathcal{E}_r^\phi)] = \text{Var}[\hat F_t(x_t;\mathcal{E}_r)] = \frac{\text{Var}[\tilde p_r(x_r)]}{\text{Var}[\tilde p_1(x_1)]}\text{Var}[\hat F_t(x_t)].
> $$
> where $\hat F_t(x_t)$ is the estimator without bootstrapping. Since $\text{Var}[\tilde p_r(x_r)] < \text{Var}[\tilde p_1(x_1)]$, the variance of bootstrapping estimator is smaller than the $\hat F_t(x_t)$.
>
> ---
> ### **W1-4, Q3: Performance on high-dimensional real-world problems is unknown.**
>
> While the current experiments are in relatively low-dimensional settings, the Ising model is a widely used benchmark with relevance to real-world physics problems. More broadly, energy-driven training is still at an early stage, and much of the existing literature focuses on validating methods on synthetic or controlled tasks. Scaling to high-dimensional real-world problems is a promising direction for future work in this field.
>
> ---
> ### **W2-1: A bit complicated notation $\tilde{p}, \tilde{w}$ and multiple kernels appear in rapid succession, which may force readers to backtrack.**
>
> Thank you for the helpful comment. We understand that the use of notations such as $\tilde{p}$ and $\tilde{w}$, along with multiple kernels introduced in close succession, can hinder readability. In the revised version, we will make an effort to briefly clarify these notations whenever they reappear, so that the exposition remains accessible without requiring readers to backtrack.
>
> ---
> ### **W2-2: Some important results (intractability of the backward kernels) are primarily deferred to the appendices.**
>
> Thank you for the feedback. We agree that the intractability of the backward kernels is important, and we will move the key parts of this analysis from the appendix to the main text in the revised version to improve clarity.
>
> ---
> ### **W3-1, Q2: No comparisons with other cutting-edge baselines (e.g., any diffusion-based schemes or generator matching)**
>
> We agree on the importance of more baselines; in response, we include results on the continuous many-well 32 benchmark comparing our method with diffusion-based samplers iDEM and PIS:
>
> **Continuous task (manywell 32):**
> |Method|Energy w1|Sample w2| # of modes found (in projection plot) |
> |-|-|-|-|
> | EGM + BS | 3.30 | 7.07 |4|
> |iDEM | 5.53 | 7.74 |4|
> | PIS | 3.39 | 6.14 |1 (mode collapsed)|
>
> Regarding Generator Matching (GM) and Flow Matching (FM), they typically assume access to samples from the target distribution. As such, they are not directly applicable to our problem formulation. We believe that the included baselines (iDEM and PIS), which also operate in energy-only settings, offer a more relevant and fair comparison and help address your concern.
>
> ---
> ### **W3-2: Some may view EGM as an incremental extension of GM.**
> While it is true that our method builds on known components such as importance sampling and generator matching, we believe that their combination in the energy-only setting with bootstrapped estimation represents a significant and broadly applicable contribution.
>
> Unlike generator matching that requires access to data, EGM operates entirely from energy evaluations, making it applicable to a wide range of domains where sampling is difficult or impossible — including discrete, continuous, and hybrid state spaces. In that sense, we view EGM not as a narrow specialization, but as a general framework for neural sampler training across arbitrary state spaces and stochastic processes (diffusions, flows, and jumps).
>
> ---
> ### **W4-1: The discussion of connection to other methods (GFlowNet) is presented, but somewhat limited (e.g., no numerical comparison).**
> Thank you for the suggestion. While our main focus is on proposing a continuous-time neural sampler for general state spaces, we agree that a deeper investigation into the connection with GFlowNets would enrich the broader context of our work. We consider this an interesting and promising direction, and we would like to explore both theoretical and numerical comparisons with GFlowNet-based methods in future work.
>
> ---
> ### **Q1: Method's efficacy depending on the choice of bootstrapping time step $\epsilon$?**
> We present a table showing how the performance of bootstrapping varies with the time gap $\epsilon = r - t > 0$ between two time steps $r, t$ (on Ising model 5 x 5, $\beta = 0.4$).
> | $\epsilon$   | Energy W_1 | Magnet W_1 |
> | --------| --------| --------|
> | 0.01 | 2.84|  0.32 |
> | 0.02 | 0.96|  0.09|
> | 0.05 | 0.84|  0.08|
> | 0.1  | 0.58|  0.08|
> | 0.2  | 1.90|  0.09|
> | 0.5  | 3.73|  0.20|
>
> As $\epsilon$ decreases, bootstrapping improves performance. However, if $\epsilon$ becomes too small, the scale of the conditional generator $F_{t|r}$ increases rapidly, leading to large fluctuations in the loss magnitude across time. This makes neural network optimization unstable due to inconsistent gradient scales. Hence, selecting a moderately small $\epsilon$ is crucial for stable training.

---

> > ### Comment · Reviewer_UQ9j · 2025-08-05
> >
> > Thank you for your response to my questions. I think the additional proposition for the bias/variance of the IS sampler will be a great addition to the paper. I think the paper is very interesting and a nice contribution to the setting where you do not have samples from the target density. I'm still a bit disappointed with the limited range of experiments, and so I will maintain my current score of borderline accept.

---

> > > ### Author Response · Authors · 2025-08-06
> > >
> > > Thank you for your thoughtful follow-up. We're pleased that the additional proposition on the bias and variance of the importance sampling estimator was useful, and we truly appreciate your recognition of the novelty of our contribution in the setting without access to target samples. Your feedback has been very valuable to us.

---

### Official Review · Reviewer_29F6 · 2025-06-23

**Clarity:** 2
**Significance:** 2
**Originality:** 2
**Rating:** 4
**Confidence:** 4

**Summary:**

This paper tries to solve the problem of sampling from distribution with given energy function (unnormalized log probability). Given a target distribution, the authors introduce a series of intermediate distributions similar to diffusion/flow matching model. They then introduce a generator function $F_{t}^{\theta}$, which is trained with the generator matching loss. To reduce the variance in the importance weight, the authors further introduce an additional bootstrapping trick. The authors evaluate their method on several toy examples and compared with the basic Gibbs Sampler.

**Questions:**

No extra questions

**Ethical Concerns:**

["NO or VERY MINOR ethics concerns only"]

**Final Justification:**

My previous concern lies in the relation between the proposed method and the previous EBM model works. The authors raise an interesting point on possible mode collapse during rebuttal, which resolved my concern. Thus, I increase my rating to 4.

**Limitations:**

Yes, the authors have adequately addressed the limitations and potential negative societal impact

**Paper Formatting Concerns:**

No paper formatting issue.

**Quality:**

2

**Strengths And Weaknesses:**

***Strength:***

This paper proposes a valid sampler and provides sufficient derivations and theoretical justifications to facilitate reader understanding.

***Weakness***

The proposed method is only evaluated on relatively simple distributions and is compared solely with the basic Gibbs sampler. In fact, the task of sampling from a given energy-function remind me of the research of Energy-Based Model (EBM), where the objective is to learn the underlying unnormalized data distribution.

A classical approach to EBM training is based on Maximum Likelihood Estimation, which involves alternating between updating the energy function and sampling from the given energy function. To accelerate the sampling process, Cooperative Training [1] introduces an auxiliary generator trained via MCMC-based teaching. Subsequent works [2, 3] extend this framework to multi-scale or multi-noise-level variants, which bear strong resemblance to the generator design adopted in this paper.

Importantly, EBM training is typically conducted on highly complex, multimodal distributions—such as natural images or 3D shapes—where the energy functions are implemented as deep neural networks. In contrast, the distributions studied in this paper are relatively simple, fixed, and analytically tractable. Given appropriate hyperparameter settings, it is plausible that existing methods from [1, 2, 3] would already be capable of addressing the tasks considered in this paper.

Therefore, I have concerns regarding the motivation and significance of the proposed approach.

[1] Cooperative Training of Descriptor and Generator Networks

[2] Learning energy-based models by cooperative diffusion recovery likelihood

[3] Improving Adversarial Energy-Based Model via Diffusion Process

---

> ### Author Rebuttal · Authors · 2025-07-31
>
> We appreciate your insightful comment and the connection to existing EBM literature. While these works are relevant, we would like to highlight the key difference in problem setting, as described below.
>
> ### **W1: It is plausible that existing methods from [1, 2, 3] would already be capable of addressing the tasks considered in this paper.**
>
> Our task is fundamentally different from the suggested works [1, 2, 3]. The suggested papers focus on energy-based "models", while our work focuses on energy-based "training". The suggested papers aim to train an energy function "from data", while our work aims to train a model "from energy functions".
>
> We hope our responses have addressed your questions and concerns. Please let us know if there is anything we may have missed or if any points require further clarification.

---

> > ### Comment · Reviewer_29F6 · 2025-08-01
> >
> > Thank the authors for their response.
> >
> > I understand the difference for energy-based "models" and energy-based "training". However, it seems to me that the task of energy-based "training" mentioned here (sampling from a given probability distribution) is a sub-task of energy-based "modeling" (learning both distribution from data and sampling from the learned probability distribution). The learned probability distribution in energy-based "modeling" is generally more complex than the ones with closed-form expressions. My concern is that **whether "sampling from a given distribution" is still challenging or it has already been solved by previous work.**

---

> ### Author Response · Authors · 2025-08-03
>
> Thank you for following up on our rebuttal and clarifying your point! We do not take it for granted. In what follows, we provide several arguments that show how the suggested energy-based models [1,2,3] do not solve energy-based training (without ground-truth data) and how our contribution is meaningful despite the suggested references.
>
> **1. Suggested works [2,3] require initialization from ground-truth data.**
>
> The suggested methods [2, 3] critically rely on the assumption of data being available. To be specific, the methods require Markov chains initialized with ground truth samples to sample from the parameterized energy-based models. Without the ground truth samples, which are a good approximation of samples from the energy-based models, the algorithms would require a prohibitive amount of computation. Hence, their performance would degrade significantly for image generation or our settings where ground truth data is unavailable.
>
> Below, we concretize our arguments for each of the suggested methods:
> - **CDRL** [2] trains a generator (the recovery-likelihood model) by running MCMC chains initialized from noise-added ground-truth data.
> - **DDAEBM** [3] trains a variational generator by aligning it to the conditional EBM via a divergence, using noise-added ground-truth data.
>
> **2. Cooperative training [1] is vulnerable to mode collapse in our setting.**
>
> Cooperative training [1] updates the generator by running short MCMC chains initialized from its own samples and maximizing the MLE objective on the resulting MCMC samples. However, if the generator misses some modes, short MCMC cannot reach them, and the model fails to correct its bias—leading to mode collapse.
>
> We empirically compare cooperative training with our method (EGM + BS) on a continuous 32-dimensional many-well energy function [5]. As shown below, our method discovers more modes and achieves lower energy, indicating better coverage of the energy landscape.
>
> |Method|Energy  $\mathcal{W}_1$|Sample $\mathcal{W}_2$| # of modes found (in projection plot) |
> |-|-|-|-|
> |EGM + BS|3.30|7.07|4|
> |Cooperative training (with fixed energy) |10.79|6.51|1 (mode collapsed)|
>
> While cooperative training achieves a lower sample $\mathcal{W}_2$ score, this does not reflect comparable performance. The low score results from mode collapse, where samples are concentrated around a single mode—a known failure case for $\mathcal{W}_2$. In contrast, our method achieves broader coverage of the distribution with substantially higher sample quality.
>
> **3. Our work is more general.**
> We also remark that our main focus is to develop a general framework that allows sampling both discrete and continuous variables for a broad class of continuous-time Markov chains (diffusion, flow, and jump), while the suggested algorithms are limited to diffusion-based or one-shot generation of continuous variables.
>
> [1] Cooperative training of descriptor and generator networks.
>
> [2] Learning energy-based models by cooperative diffusion recovery likelihood.
>
> [3] Improving adversarial energy-based model via diffusion process.

---

> > ### Comment · Reviewer_29F6 · 2025-08-04
> >
> > Thank the authors for their response.
> >
> > I think the authors have made an interesting pointing regarding the possible mode collapse with the previous work. In EBM training, the existence of ground truth samples will guide the learned energy function towards learning all the modes, which further guides the learning of sampler/initializer. If purely energy function is given, cooperative learning/adversarial based algorithm might have mode collapse if the initialization is bad or lack of exploration in MCMC (I think the observed mode collapse in the new table might be also due to the lack of tuning for MCMC hyper-parameters). In this sense, I agree the proposed method is solving a related but not exactly the same problem. Thus, I would like to increase my rating to 4. And I think it might be worth adding some discussion regarding this difference in the paper.

---

> > > ### Author Response · Authors · 2025-08-04
> > >
> > > Thank you for your thoughtful follow-up and for increasing the rating. We appreciate your recognition of the distinction in problem settings. We will make sure to include a discussion on this point in the future revision. We also acknowledge the importance of careful MCMC hyper-parameter tuning and will do more extensive tuning in the future manuscript.

---

### Official Review · Reviewer_meoi · 2025-07-01

**Clarity:** 3
**Significance:** 2
**Originality:** 2
**Rating:** 3
**Confidence:** 4

**Summary:**

Setting
--

This is a method to train a continuous time Markov process in order to draw samples from the Boltzmann model:

p_{target} \propto \exp ( - E(x) )
with known energy E(x).

The goal is to have a single approach to generate  from continuous, discrete, mixed modalities. The method is demonstrated concretely to have as special cases diffusion, flow, and masked diffusion models.

The work builds on the recent paper of Holderrieth et al, ``Generator Matching'' [16], which
already unifies  diffusion models, flow matching, and discrete diffusion models;
explores jump processes, and allows for multimodal generation (continuous, discrete, mixed); but is data driven (vs the Boltzmann model in this submission).

As noted by [16], diffusion, flow, and jump processes are instances of continuous time Markov processes with a linear parametrization. The authors use F_t(x) to define the linear parametrization of a given infinitesimal generator which goes from noisy to clean.    The reparametrization trick (3) relates F_t to the conditional generator with parametrization F_{t|1}^{x_1}. The loss (4) gives a loss from the true F_t to the model F_t^{\theta}.

Approaches
--

A first approach is Energy-based Generator matching, where an approximation to F_t (x_t) is obtained with a self normalized importance weighted estimator with respect to distribution q_{1|t}, with the weights being a function of the unnormalized \tilde p, and the normalization constant being computed as in (8). Eq. (4) then becomes eq. (9).

The challenge is in the design of q_{1|t}. A naive approach with high variance is to set it to p_{t|1}.  A second ``bootstrapping'' approch with lower variance is to use a nearby time r to t, under the weaker assumption  p_{r|t} \approx p_{t|r}.  Eq. (14) gives the equation for L_t as a function of L_{t|r}^{x_r} and p_{r|t}.

Then p_{r|t} is estimated using an importance sampler wrt q_{r|t}, which is chosen to be approximated by p_{t|r}, with lower variance.

This leads to another challenge: that the weights require \tilde p_r (x_r). The estimator in eq. 18 has a similar high variance problem to the first approach, in that it approximates q_{1|r} \propto p_{r|1}.  The model for \tilde p_{r} is trained by noised energy matching.

**Questions:**

Is there a reason for eq. 5 being included? Is it used?

**Ethical Concerns:**

["NO or VERY MINOR ethics concerns only"]

**Final Justification:**

I am maintaining my score due to continued concerns regarding the theoretical contribution.

**Limitations:**

yes

**Quality:**

2

**Strengths And Weaknesses:**

Strengths:
--

the idea is original (notwithstanding the dependence on [16], which is clearly and appropriately cited), the method is clearly explained, and performance on discrete and mixed continuous-discrete examples outperforms a baseline classical Gibbs sampler.



Main theory limitation:
--

It is not obvious to me that eq. 18 *provably* gets around the high variance problem, since it again makes the approximation  q_{1|r} \propto p_{r|1} for a large jump in time from r to 1 (assuming t is close to r, to allow for small variance in that step).

So it seems you have just shifted the same problem to another part of the reasoning.

It is also not at all clear why, or if, the fact that the surrogate loss is used (eq. 20) to get a parametric model for \tilde p_r solves the variance problem in a meaningful way since the loss is with respect to (19), which has high variance.  This would limit the accuracy with which one could accurately compute \tilde p_r .

Can you give a more formal arguemnt for why ``bootstrapping'' should be expected to guarantee lower variance? I understand that empirically it appears to do better, but I wonder if there is a theoretical argument to be made?


Main experiment limitation:
--

The evaluation was performed only on discrete (Ising) and joint discrete-continuous tasks, however the method also applies for the continuous setting. In order to evaluate the efficacy of the method, and in particular the issues that arise in eq. 18 and following, it is reasonable and necessary to benchmark the method against established methods (including those cited by the authors) for sampling from purely *continuous* EBMs (parallel tempering/SMC samplers/SOTA diffusion samplers).

---

> ### Author Rebuttal · Authors · 2025-07-31
>
> Thank you for your thoughtful and constructive review. We sincerely appreciate your recognition of the originality of our method. Please find our detailed responses to your comments below.
>
> ### **W1: Energy estimator still has high variance, so why is bootstrapping better?**
>
> Thank you for raising this important concern. Our bootstrapping helps training through a trade-off between approximation bias and estimation variance. That is, our algorithm replaces the high variance Monte Carlo approximation of intermediate energy $\mathcal{E}_r$ with a biased surrogate model $\mathcal{E}_r^\phi$. Similar to actor-critic training in reinforcement learning, this trade-off helps stabilize the training of the marginal generator by replacing the high-variance (unbiased) target with a low-variance (biased) target.
>
> To further alleviate your concern, we formally prove that the bootstrapping estimator achieves a lower variance compared to the original estimator.
>
> **Proposition.** Consider a fully trained surrogate energy model $\mathcal{E}_r^{\phi}(x_r) = \mathbb{E}[\hat{\mathcal{E}}_r(x_r)]$, where $\hat{\mathcal{E}}_r(x_r)$ is a biased energy estimator (from eq. 20). Let $\hat F_t(x_t; \mathcal{E}_r)$ be the bootstrapping estimator using $\mathcal{E}_r$ as the intermediate energy function:
> $$
> \hat F_t(x_t;\mathcal E_r) = \frac{\sum\_i \exp (-\mathcal{E}\_r(x_r^{(i)})) F\_{t|r}\^{x_r^{(i)}}(x)}{\sum\_i \exp (-\mathcal{E}\_r(x_r^{(i)}))}.
> $$
>
> Then, the bias and the variance bound of the bootstrapping estimator with surrogate model $\hat F_t(x_t; \mathcal{E}_r^\phi)$ are given by:
> $$
> \text{Bias}[\hat F_t(x_t;\mathcal{E}_r^\phi)] = \text{Bias}[\hat F_t(x_t;\mathcal{E}_r)] +  O\left(\frac{1}{K^2}\right),
> $$
> $$
> \text{Var}[\hat F_t(x_t;\mathcal{E}_r^\phi)] = \text{Var}[\hat F_t(x_t;\mathcal{E}_r)] = \frac{\text{Var}[\tilde p_r(x_r)]}{\text{Var}[\tilde p_1(x_1)]}\text{Var}[\hat F_t(x_t)].
> $$
> where $\hat F_t(x_t)$ is the estimator without bootstrapping. Since $\text{Var}[\tilde p_r(x_r)] < \text{Var}[\tilde p_1(x_1)]$, the variance of bootstrapping estimator is smaller than the $\hat F_t(x_t)$.
>
> **proof sketch)** The bias of the energy estimator $\hat{\mathcal{E}}_r$ is given as follows (refer to Corollary 3.2 of BNEM [25]):
> $$
> \text{Bias}[\hat{\mathcal{E}_r}(x_r)] = \frac{\text{Var}[\tilde p_1(x_1)]}{2p_r^2(x_r)K} + O\left(\frac{1}{K^2}\right)
> $$
> Thus, we can approximate the surrogate energy as,
> $$
> \mathcal{E}_r^\phi(x_r) = \mathbb{E}[\hat{\mathcal{E}}_r(x_r)]=\mathcal{E}_r(x_r)+\underbrace{\frac{\text{Var}[\tilde p_1(x_1)]}{2p_r^2(x_r)K}}\_{:=b(x_r)}.
> $$
>
> Plugging the equation into the $\hat{F}_t(x_t;\mathcal{E}_r^\phi)$, we get,
>
> $$
> \hat F_t(x_t;\mathcal{E}_r^\phi) = \frac{\sum_i \exp (-\mathcal{E}\_r^\phi(x_r\^{(i)})) F\_{t|r}^{x_r\^{(i)}}(x_t)}{\sum_i \exp (-\mathcal{E}_r^\phi(x_r\^{(i)}))} = \frac{\sum_i \exp(-\mathcal{E}_r(x_r\^{(i)}) - b(x_r\^{(i)}))F\_{t|r}^{x_r\^{(i)}}(x_t)}{\sum_i \exp(-\mathcal{E}_r(x_r\^{(i)}) - b(x_r\^{(i)}))}
> $$
>
> Here, $b(x_r^{(i)})$ is close to 0 and concentrated to:
>
> $$
> m_b= b(x_t)=\frac{\text{Var}[\tilde p_1(x_1)]}{2p_r^2(x_t)K},
> $$ when we choose $t$ close to $r$ and large $K$. Then, by applying first-order Taylor expansion to the $\hat F_t(x_t;\mathcal{E}_r^\phi)$ and approximation $b(x_r\^{(i)}) \approx m_b$, we obtain:
>
> $$
> \hat F_t(x_t;\mathcal{E}_r^\phi) \approx (1 - m_b^2)\frac{\sum_i \exp(-\mathcal{E}_r(x_r\^{(i)}))F\_{t|r}\^{x_r\^{(i)}}(x_t)}{\sum_i \exp(-\mathcal{E}_r(x_r\^{(i)})))} = (1 - m_b^2) \hat F_t(x_t;\mathcal{E}_r)
> $$
>
> Therefore, the bias of the bootstrapping estimator $\hat F_t(x_t;\mathcal{E}_r^\phi)$ is:
>
> $$
> \text{Bias}[\hat F_t(x_t;\mathcal{E}_r^\phi)] = \mathbb{E}[\hat F_t(x_t;\mathcal{E}_r^\phi)] - F_t(x_t) = \text{Bias}[\hat F_t(x_t;\mathcal{E}_r)] - m_b^2 \left(F_t(x_t) + \text{Bias}[\hat F_t(x_t;\mathcal{E}_r)]\right) = \text{Bias}[\hat F_t(x_t;\mathcal{E}_r)] +  O\left(\frac{1}{K^2}\right)
> $$
> Similarly, the variance of the bootstrapping estimator is:
> $$
> \text{Var}[\hat F_t(x_t;\mathcal{E}_r^\phi)] = (1-m_b^2)^2 \text{Var}[\hat F_t(x_t;\mathcal{E}_r)] \approx \text{Var}[\hat F_t(x_t;\mathcal{E}_r)]
> $$
> since $m_b < 1$ for sufficiently large $K$.
>
>
> Also, the variance of the bootstrapping estimator $\hat F_t(x_t;\mathcal{E}_r)$ with true energy is given by:
> $$
> \text{Var}[\hat F_t(x_t;\mathcal{E}_r)]=\frac{4\text{Var}[\tilde p_r(x_r)]}{p_t^2(x_t)K}(1 + \lVert F_t(x_t)\rVert)^2,
> $$
>
> Consequently,
> $$
> \text{Var}[\hat F_t(x_t;\mathcal{E}_r^\phi)] \approx \text{Var}[\hat F_t(x_t;\mathcal{E}_r)] = \frac{\text{Var}[\tilde p_r(x_r)]}{\text{Var}[\tilde p_1(x_1)]}\text{Var}[\hat F_t(x_t)].
> $$
> Because $\text{Var}[\tilde p_r(x_r)] \ll \text{Var}[\tilde p_1(x_1)]$ is very small, the variance of the bootstrapping estimator is smaller than the $\hat{F}_t(x_t)$. Consequently, with an fully trained energy model, we **reduce the variance of the SNIS generator estimation**.
>
> [25] OuYang et al. BNEM: A boltzmann sampler based on bootstrapped noised energy matching, ICLR 2025.
>
> ---
> ### **W2: Benchmark on the continuous task is missing.**
>
> Thank you for the suggestion. To evaluate EGM in a continuous setting, we added results on the many-well 32 benchmark, comparing against iDEM and PIS, a diffusion-based baseline.
>
>
> **Continuous task (manywell 32):**
> |Method|Energy w1|Sample w2| # of modes found (in projection plot) |
> |-|-|-|-|
> | EGM + BS | 3.30 | 7.07 |4|
> | iDEM | 5.53 | 7.74 | 4|
> | PIS | 3.39 | 6.14 | 1 (mode collapsed) |
>
> As the results show, EGM matches or slightly outperforms existing diffusion-based samplers (iDEM, PIS) in the continuous setting.
>
> ---
> ### **Q1: Is there a reason for Eq. 5 being included?**
> Eq. 5 was included to illustrate how to solve the intractability of marginal generator matching for data-driven training. However, since it is not used later in the paper, we will remove it in the revision.
>
> ---
> We hope our responses have addressed your questions and concerns. Please let us know if there is anything we may have missed or if any points require further clarification.

---

> ### Comment · Reviewer_meoi · 2025-08-07
> **Update**
>
> Thank you for your reply. On the theory side, unfortunately the reply does not address my main concern.
>
> * Eq. (18) makes the approximation q_{1|r} \propto p_{r|1} for a large jump in time from r to 1 (assuming t is close to r, to allow for small variance in that step).
>
> Previously, in line 135, you stated that approximating q_{1|t} \propto p(t|1) leads to "high variance in the importance weight." So you then make a smaller step to get p_{r|t} which you approximate by p(t|r).  When t and r are close, the variance of these new weights are small.
>
> **But you are still left with the problem of q_{1|r} \propto p_{r|1}.**
>
> You address this problem by introducing bias and reducing variance using (20) - I don't dispute this argument. But it does not address the end-to-end performance of the procedure as a whole, or the reason for introducing the intermediate "r".  For example, why  not introduce the same eq. 20 bias/variance trick earlier for q_{1|t} \propto p(t|1) ? Why the intermediate step at r?
>
> Re the experiments: thank you for the new experiments - these strengthen the paper.
>
> Due to the  continued theory concern, I will maintain my score.

---

> ### Author Response · Authors · 2025-08-08
>
> Thank you for following up and restating your concern. Below are detailed responses to your concerns.
>
> ---
> ### **Q1. Why introduce the intermediate step $r$ instead of applying the Eq. 20 bias-variance trick directly to generator estimation?**
> We already apply the analogue of Eq. 20’s bias–variance trade-off to generator estimation via self-normalized importance sampling (SNIS; Eq. 9). However, when temporal gap $t\rightarrow 1$ is large, direct generator estimation remains inaccurate because the estimator depends on two separate sources of variability: the unnormalized density $\tilde p_1(x_1)$ and the conditional generator $F_{t|1}^{x_1}(x_t)$. The combined effect of these two factors leads to high variance that SNIS alone cannot suppress. Bootstrapping addresses this by splitting the high-variance $t \rightarrow 1$ estimation into two lower-variance subproblems, at the cost of a bias from the surrogate energy.
>
> ---
> ### **Q2. But you are still left with the problem of $q_{1|r}\propto p_{r|1}$. How does energy estimation with $q_{1|r}$ address the high-variance problem?**
>
> As we noted, introducing an intermediate step $r$ through bootstrapping decomposes the high-variance generator estimation from $t\rightarrow 1$ into two subproblems: $t\rightarrow r$ and $r\rightarrow 1$. As you have acknowledged, this effectively reduces the variance in the first step $t\rightarrow r$. For the second step $r\rightarrow 1$ involving energy estimation, we outline several reasons why this estimation problem has lower variance.
>
> **1. Energy estimation at time $r$ is easier than the estimation at time $t$.**
>
> We set $\epsilon$ to a modest value (e.g., $\epsilon\approx 0.1$) so that $r$ lies closer to $1$ than $t$. This ensures that the $t\rightarrow r$ estimation retains low variance, while the $r\rightarrow 1$ estimation also benefits from a smaller temporal gap. By narrowing this gap relative to $t\rightarrow 1$, the variance of the energy estimator at $r$ is reduced.
>
> **2. Energy estimator $\hat{\mathcal{E}}_t$ has lower variance than the marginal generator estimator $\hat F_t$.**
>
> Even for the same temporal gap, the energy estimator $\hat{\mathcal{E}}_t$ exhibits lower variance than the generator estimator $\hat F_t$. Energy estimation depends only on unnormalized density $\tilde{p}_1(x_1)$, while generator estimation additionally involves the conditional generator $F\_{t|1}\^{x_1}(x_t)$. This extra source of variability makes the generator estimator more sensitive to the temporal gap. As a result, the variance of $\hat F_t$ is larger than that of $\hat{\mathcal{E}}_t$; we provide a formal argument for this inequality below.
>
> **Proposition.** Let $\tilde p_1(x_1)$ be an unnormalized density and $F_{t|1}^{x_1}(x_t)$ a conditional generator, evaluated for $x_1 \sim q_{1|t}$. Suppose $\tilde p_1(x_1)$ and $\big\lVert \tilde p_1(x_1)F\_{t|1}\^{x_1}(x_t)\big\rVert$ are sub-Gaussian. Then the variances of the energy estimator and the generator estimator satisfy:
> $$
> \text{Var}[\hat{\mathcal{E}}_t(x_t)]=\frac{\text{Var}[p_1(x_1)]}{p_t^2(x_t)K}+O\left(\frac{1}{K^2}\right),
> $$
> $$
> \text{Var}[\hat F_t(x_t)]=\frac{4\text{Var}[p_1(x_1)]}{p_t^2(x_t)K}\left(1+\lVert F_t(x_t)\rVert \right)^2+O\left(\frac{1}{K^2}\right),
> $$
> where $K$ is the sample size.
>
> A derivation of the variance for the generator estimator is provided in our response to Reviewer Vwnf (item W3-1), and that for the energy estimator can be found in Corollary 3.2 of [1]. Comparing the leading terms, $\text{Var}[\hat F_t]$ is strictly larger than $\text{Var}[\hat{\mathcal{E}}_t]$ since $4\left(1+\lVert F_t(x_t)\rVert \right)^2>1$.
>
> **3. The accuracy of energy estimates can be further improved.**
>
> Unlike generator estimation, the accuracy of energy estimation can be enhanced using established techniques. In our method, the forward-looking parametrization evaluates the target energy at intermediate states, yielding an effective approximation to the true intermediate energy. Moreover, as shown in [1], bootstrapping can be applied again to energy estimation, which provably reduces both variance and bias (Proposition 3.4 in [1]).
>
> **4. We replace the MC energy estimator with variance to the learned biased surrogate.**
>
> To further mitigate variance in energy estimation, we replace the raw Monte Carlo energy estimator with a learned biased surrogate $\mathcal{E}_r^\phi$ within the bootstrapping. As discussed in our previous response, combining bootstrapping with the biased learned surrogate effectively reduces the variance of generator estimation, supporting our claim that **the overall bootstrapping framework achieves a lower-variance estimator**.
>
> ---
> We hope these clarifications address your concerns. If any part remains unclear, please let us know during the remaining discussion period, and we will be glad to elaborate further. Thank you again for your thoughtful feedback.
>
> [1] OuYang et al. BNEM: A boltzmann sampler based on bootstrapped noised energy matching, ICLR 2025.

---

### Official Review · Reviewer_Vwnf · 2025-07-03

**Clarity:** 3
**Significance:** 2
**Originality:** 2
**Rating:** 4
**Confidence:** 3

**Summary:**

This paper introduces Energy-based Generator Matching (EGM), which extends the generator matching framework to scenarios where only energy functions are available rather than pre-computed equilibrium samples. The authors propose using self-normalized importance sampling to estimate the marginal generator in the absence of training data, along with a bootstrapping technique that leverages intermediate time steps with learned energy estimators to reduce variance in importance weights. EGM aims to provide a unified framework for training neural samplers across arbitrary continuous-time Markov processes (diffusion, flow, and jump processes) on general state spaces including continuous, discrete, and mixed modalities. The method is validated on several tasks including the 2D Ising model, Gaussian-Bernoulli RBM, and joint discrete-continuous sampling problems.

**Questions:**

- Missing Baseline Comparisons: Why weren't key energy-based neural samplers like iDEM, DDS, or PIS compared? These methods also work in energy-only settings and would provide much more meaningful baselines than Gibbs sampling alone. Can you provide experimental comparisons showing where and why EGM outperforms these existing approaches?

- The paper acknowledges training instability and bias from self-normalized importance sampling but provides limited analysis. Can you provide theoretical or empirical analysis of: (a) convergence guarantees for the bi-level training scheme, (b) bias bounds for the SNIS estimator, and (c) conditions under which the method is expected to be stable?

- Scalability Demonstration: The experiments are limited to moderate dimensions (≤100). Can you demonstrate the method's effectiveness on higher-dimensional problems where the advantages over traditional MCMC would be more compelling? What are the computational scaling properties compared to existing energy-based neural samplers?

- Relationship to Existing Methods: When examining your estimator in Equation (9) and comparing to iDEM's approach, the techniques appear conceptually similar (both use importance sampling with energy-based weights). Can you clearly articulate the technical differences and when EGM would be preferred over existing methods?
- While Theorem 1 justifies the bootstrapping theoretically, can you provide more rigorous analysis of when and why it reduces variance? Under what conditions might bootstrapping fail or introduce additional bias through the learned energy estimator?

**Ethical Concerns:**

["NO or VERY MINOR ethics concerns only"]

**Final Justification:**

I have initial doubts on some points of the work, which has been answered by the authors in their rebuttal. Therefore I recommend weak acceptance to this work.

**Limitations:**

No limitations are discussed in the main text.

**Paper Formatting Concerns:**

No.

**Quality:**

2

**Strengths And Weaknesses:**

# Strengths
The paper tackles an important problem since many real-world applications naturally provide energy functions rather than equilibrium samples. I also appreciate the theoretical contribution of extending generator matching to energy-only settings, which addresses a practically important limitation. The unified framework supporting different Markov process types across general state spaces is conceptually valuable, and the mathematical development appears sound. The bootstrapping technique with Theorem 1 providing theoretical justification for using intermediate time steps is a clever contribution.

# Weaknesses
That being said, I have serious concerns mostly about the experimental evaluation of this paper.

- The authors of this paper fail to compare against key existing energy-based neural samplers. For example, with discrete tasks, LEAPS should have been compared, and for energy-based sampling, several relevant baselines exist.

- Moreover, the experimental scope is insufficient for the claims made. The largest experiments are on 100-dimensional Ising models and 20-dimensional mixed problems, which doesn't demonstrate scalability to the high-dimensional problems where such methods would be most valuable. The authors acknowledge training instability issues but provide only ad-hoc solutions (exponential moving averages) without thorough analysis.
- The technical quality is solid but not exceptional. The bias introduced by self-normalized importance sampling and potential proposal-posterior mismatch is mentioned but not rigorously analyzed. I think the paper would benefit from theoretical guarantees about convergence properties and bias bounds. The relationship between EGM and existing energy-based samplers like iDEM is unclear. For example, Equation (9) appears conceptually similar to existing importance sampling approaches in the energy-based sampling literature.

---

> ### Author Rebuttal · Authors · 2025-07-31
>
> Thank you for your thoughtful and encouraging review. We appreciate your recognition of our theoretical contributions. Please find our detailed responses below.
>
> ### **W1, Q1: Missing baseline (e.g., iDEM, PIS, DDS, and LEAPS)**
> We agree on the importance of more baselines; however, we would also like to emphasize that EGM is not directly comparable with the baselines. EGM is a general framework applicable to discrete, continuous, or even discrete-continuous hybrid tasks, while the baselines focus on one of the discrete or continuous tasks.
>
> Nevertheless, we provide a comparison with the baselines. For continuous tasks, we compare with iDEM and PIS on the many-well 32 [6]. We omit DDS from the comparison since it performs similarly to PIS. For discrete tasks, we compare with LEAPS on the 15 x 15 Ising model.
>
> **Continuous task (manywell 32):**
> |Method|Energy w1|Sample w2| # of modes found (in projection plot) |
> |-|-|-|-|
> | EGM + BS | 3.30| 7.07|4|
> | iDEM |5.53|7.74|4|
> | PIS | 3.39 | 6.14 |1 (mode collapsed)|
>
> As the results show, EGM matches or slightly outperforms existing diffusion-based samplers (iDEM, PIS) in the continuous setting.
>
> **Discrete task (Ising 15 x 15, $\beta = 0.28$):**
> |Method|Energy w1|Magnet w1|
> |-|-|-|
> |EGM| 0.89 | 0.06 |
> | EGM + BS | 0.65 | 0.04 |
> | LEAPS| 0.68 | 0.01 |
> | LEAPS + MCMC | 0.49 | 0.01 |
>
> For this case, LEAPS is on par with EGM+BS (bootstrapping). However, one can also observe that LEAPS + MCMC is better than EGM+BS at the cost of using more energy calls (45B vs. 85B).
>
> ---
> ### **W2-1, Q3: Scalability to high-dimensional problems?**
> Please refer to our response to W1, Q1, where we present results on a higher-dimensional (d = 225) Ising model to support the scalability of EGM.
>
> Additionally, our primary goal is to expand the design space (e.g., probability path, Markov transition type) and the application of energy-driven training (e.g., multi-modal case), rather than scaling a method to higher dimensions than previous research. Scaling to higher-dimensional tasks remains a challenge for all energy-driven training algorithms.
>
> ---
> ### **W2-2, Q2-3: Analysis on training instability is missing.**
>
> While the theoretical analyses on stability are outside of the scope of our paper, to alleviate your concern, we provide an empirical analysis of the training stability.
>
> Applying EMA to the energy model can stabilize the training, and we provide a result with and without EMA (on Ising 10 x 10, $\beta=0.4$). All the other hyperparameters are fixed.
>
> | Metric         | Energy w1  | Magnet w1 |
> | --------       | ---------  | --------  |
> | With EMA       | 2.51±0.16  | 0.24±0.01 |
> | Without EMA    | 10.08±8.65 | 0.45±0.13 |
>
> Using EMA on the energy-model parameters reduces variance in energy w1 and magnetization score. EMA addresses the moving‑target issue when the generator is regressed on a learned energy model. It is a well‑established technique in reinforcement learning for stabilizing value‑function updates.
>
> ---
> ### **W3-1, Q2-2: Lack of bias bounds introduced by self-normalized importance sampling.**
> We show that the SNIS estimator's error decays as $O(1/\sqrt{K})$, and its bias and variance decay as $O(1/K)$, where $K$ is the sample size. Below, we sketch the proof, and the full rigorous derivation will appear in the revised version.
>
> **Proposition.** Consider an unnormalized density $\tilde p_1(x_{1})$ and a conditional generator $F_{t|1}^{x_1}(x_{t})$ evaluated on a sample $x_1 \sim q_{1|t}$. Suppose $\tilde p_1(x_1)$ and $\Big\lVert \tilde p_1(x_1) F_{t|1}^{x_1}(x_t) \Big\rVert$ are sub-Gaussian. Then there exists a constant $c(x_t)$ such that with probability at least $1-\delta$,
> $$
> \left\lVert \hat F_t(x_t) - F_t(x_t) \right\rVert \leq c(x_t) \sqrt{\frac{\log (2/\delta)}{K}},
> $$
> where $\hat F_t(x_t)$ denotes the SNIS estimator from eq. 9 using $K$ samples $x_1^{(1)}, \\ldots, x_1^{(K)}\sim q_{1|t}$:
> $$
> \hat F_t(x_t) = \frac{\sum_{i=1}^K \tilde p_1(x_1^{(i)})F_{t|1}^{x_1^{(i)}}(x_t)}{\sum_{i=1}^K \tilde p_1(x_1^{(i)})}.
> $$
> Moreover, its bias and variance are expressed as:
> $$
> \text{Bias}[\hat F_t] = \frac{1}{Kp_t(x_t)^2} \left(- \text{Cov}\left[\tilde p_1 F_{t|1}, \tilde p_1\right] + F_t(x_t)\text{Var}[\tilde p_1] \right) + O\left(\frac{1}{K^2}\right),
> $$
> $$
> \text{Var}[\hat F_t] = \frac{4\text{Var}[\tilde p_1(x_1)]}{p_t^2(x_t)K}(1 + \lVert F_t(x_t)\rVert)^2,
> $$
>
> **proof sketch)** By Hoeffding's inequality for sub-Gaussian random variables, there exists a constant $C$ such that
> $$
> \left| \frac{1}{K} \sum_{i=1}^{K} \tilde p_1(x_1^{(i)}) - \tilde p_t(x_t) \right| \leq C \sqrt{\frac{\log (2/\delta) }{K}},
> $$
> $$
> \left\lVert \frac{1}{K} \sum_{i=1}^{K} \tilde p_1(x_1^{(i)}) F_{t|1}^{x_1^{(i)}}(x_t) - \tilde p_t(x_t)F_t(x_t) \right\rVert \leq C \sqrt{\frac{\log (2/\delta) }{K}},
> $$
> with $1 - \delta$ probability. (Here, $C = \sqrt{\text{Var}[\tilde p_1(x_1)]}$ is a possible choice.)
>
> Since we have bounds for both numerator and denominator of $\hat F_t$, we can also bound the error of $\hat F_t$ to $F_t$. The result is as follows:
>
> $$
> \left\lVert \hat F_t(x_t) - F_t(x_t) \right\rVert \leq \frac{2C}{p_t(x_t)}(1 + \lVert F_t(x_t)\rVert)\sqrt{\frac{\log(2/\delta)}{K}}=c(x_t)\sqrt{\frac{\log(2/\delta)}{K}}.
> $$
> Now, let $\hat{A}$ and $\hat{B}$ denote the numerator and the denominator of $\hat F_t$, respectively. We also let $\mathbb{E}[\hat{A}] = A$ and  $\mathbb{E}[\hat{B}] = B$. For sufficiently large $K$, the Taylor expansion is expressed as:
> $$
> \hat F_t =\frac{A}{B}+\frac{1}{B}(\hat{A}-A) - \frac{A}{B^2}(\hat{B}-B)-\frac{1}{B^2}(\hat{A} - A)(\hat{B} - B) + \frac{A}{B^3} (\hat{B} - B)^2 + O\left(\frac{1}{K^2}\right).
> $$
> To derive the final equation for the bias term, one can express the bias term as follows:
> $$
> \text{Bias}[\hat F_t] = \mathbb{E}[\hat F_t] - \frac{A}{B}=-\frac{1}{B^2}\text{Cov}[\hat{A}, \hat{B}]+ \frac{A}{B^3}\text{Var}[\hat{B}].
> $$
> Since $\text{Cov}[\hat{A}, \hat{B}] = \text{Cov}\left[\tilde p_1 F_{t|1}, \tilde p_1\right] / K$ and $\text{Var}[\hat{B}]=\text{Var}[\tilde p_1]/K$, one obtains the conclusion for $\text{Bias}[\hat F_t]$.
>
> To derive the final equation for the variance term, one can combine the sub-Gaussianity of $\hat F_t$ with the error bound on $\left\lVert \hat F_t(x_t) - F_t(x_t) \right\rVert$.
>
> ---
> ### **W3-2, Q4: Relationship to iDEM is unclear.**
>
> While both iDEM and EGM use importance sampling with energy-based weights, they rely on different score identities with distinct applicability.
>
> iDEM is based on the target score identity (eq. 5 in [37]), which requires access to the target energy gradient and is restricted to the VE probability path. In contrast, EGM adopts the denoising score identity (eq. 2 in [37]), which only requires the conditional generator and applies broadly to general Markov models and state spaces.
>
> The two approaches also differ in variance behavior across time. As shown in [37], combining both estimators can reduce variance in data-driven training. Extending this idea to energy-driven training is a promising direction for future work.
>
> In short, EGM complements and extends iDEM by supporting more general settings.
>
> [37] De Bortoli, Valentin et al. Target Score Matching, arXiv:2402.08667, 2024.
>
> ---
> ### **Q2-1: Can you provide a theoretical or empirical analysis of convergence guarantees?**
>
> While a formal convergence analysis is beyond the scope of this work, to address your concern, we instead provide an empirical study on the convergence behavior of both EGM and the proposed bootstrapping method.
>
> We report the number of training epochs required for convergence. We define convergence as the point at which the evaluation metric (e.g., energy Wasserstein-1) shows no further significant decrease. (on Ising 10 x 10, $\beta = 0.4$)
>
> | Method | # of training steps|
> |-|-|
> |EGM| 10K |
> |EGM + BS| 60K |
>
> ---
> ### **Q5: Can you provide a more rigorous analysis of when and why bootstrapping reduces variance?**
>
> We prove that bootstrapping reduces variance compared to the non-bootstrapping estimator. We first show that the bias and variance of the bootstrapping estimator with a surrogate energy are close to those of the estimator using the true intermediate energy (i.e., the effect of energy estimation bias is negligible). We then show that bootstrapping with the true energy achieves lower variance than the non-bootstrapping estimator.
>
> This result suggests that, when the surrogate energy is well-trained, bootstrapping effectively reduces the variance of generator estimation.
>
> We state the proposition below. A full proof is provided in our response to Reviewer meoi's comment (W1) and will be included in the revised manuscript.
>
> **Proposition.** Consider an fully trained surrogate energy model $\mathcal{E}_r^{\phi}(x_r) = \mathbb{E}[\hat{\mathcal{E}}_r(x_r)]$, where $\hat{\mathcal{E}}_r(x_r)$ is a biased energy estimator (from eq. 20). Let $\hat F_t(x_t; \mathcal{E}_r)$ be the bootstrapping estimator using $\mathcal{E}_r$ as the intermediate energy function:
> $$
> \hat F_t(x_t;\mathcal E_r) = \frac{\sum\_i \exp (-\mathcal{E}\_r(x_r^{(i)})) F\_{t|r}\^{x_r^{(i)}}(x)}{\sum\_i \exp (-\mathcal{E}\_r(x_r^{(i)}))}.
> $$
>
> Then, the bias and the variance bound of the bootstrapping estimator with surrogate model $\hat F_t(x_t; \mathcal{E}_r^\phi)$ are given by:
> $$
> \text{Bias}[\hat F_t(x_t;\mathcal{E}_r^\phi)] = \text{Bias}[\hat F_t(x_t;\mathcal{E}_r)] +  O\left(\frac{1}{K^2}\right),
> $$
> $$
> \text{Var}[\hat F_t(x_t;\mathcal{E}_r^\phi)] = \text{Var}[\hat F_t(x_t;\mathcal{E}_r)] = \frac{\text{Var}[\tilde p_r(x_r)]}{\text{Var}[\tilde p_1(x_1)]}\text{Var}[\hat F_t(x_t)].
> $$
> where $\hat F_t(x_t)$ is the estimator without bootstrapping. Since $\text{Var}[\tilde p_r(x_r)] < \text{Var}[\tilde p_1(x_1)]$, the variance of bootstrapping estimator is smaller than the $\hat F_t(x_t)$.
>
> ---
> We hope our responses have addressed your questions and concerns. Please let us know if there is anything we may have missed or if any points require further clarification.

---

> > ### Comment · Reviewer_Vwnf · 2025-08-05
> >
> > Thank you for the thorough explanation. In light of the new content from the rebuttal, I am raising my score leaning towards acceptance.

---

> ### Author Response · Authors · 2025-08-05
>
> We’re grateful that our rebuttal helped clarify your concerns. Thank you for your constructive feedback and positive reassessment.

---

### Note · Authors · 2025-08-13

Dear AC and reviewers,

We sincerely appreciate your valuable time and effort spent reviewing our manuscript. Your comments have greatly helped us refine and strengthen the work.

As highlighted by the reviewers, our work addresses the practically important setting of energy-driven training, offering a unified framework that accommodates various types of Markov processes across arbitrary state spaces (reviewers Vwnf, UQ9j). Our method was recognized as original and creative (Vwnf, meoi, UQ9j), and was praised for its theoretical soundness (Vwnf, 29F6, UQ9j) and clear presentation (meoi, 29F6, UQ9j).

The main points for improvement were the need for more extensive empirical validation (Vwnf, meoi, UQ9j) and theoretical analyses clarifying the error bounds and the advantages of our method (Vwnf, UQ9j). In response, we added experiments with additional baselines (LEAPS, iDEM, PIS) across diverse settings, and provided both a formal error bound and a theoretical argument supporting our approach. We are encouraged that reviewers expressed satisfaction that these updates resolve their concerns, and we will incorporate them into the revised manuscript.

Reviewer meoi also encouraged us to elaborate further on the theoretical benefit of our bootstrapping method. While our work primarily focuses on algorithmic design and already includes empirical evidence of this benefit, we have now supplemented it with theoretical results and a more detailed explanation showing how bootstrapping decomposes a high-variance estimation problem into two lower-variance subproblems, thereby reducing the overall estimation variance, at the cost of introducing model learning bias. This reflects the common bias–variance trade-off in RL, and we believe this additional clarification further strengthens the paper.

We believe these updates further demonstrate the benefits and broad applicability of the proposed EGM framework to the NeurIPS community.

Thank you very much.

Sincerely,

The Authors

---

### Decision · Program_Chairs · 2025-09-17

**Decision:**

Accept (poster)

**Comment:**

This paper adapts generator matching (GM) to the case where the target distribution is given as a normalized positive function, using importance sampling and bootstrapping techniques to reduce variance of the important sampling estimator.

Most reviewers found the problem tackled by the paper important and believe this paper introduces a valuable theoretical advancement in generalization of GM to energy functions.
The main criticism in the reviews was about: 1) missing comparisons against relevant baselines; 2) theory questions regarding high variance, missing convergence proof and bias bounds, and connection to other energy based sampler; 3) method not tested in scale, experiments include relatively simple distributions; and 4) method somewhat incremental over GM. Points 1 and 2 were addressed to reasonable extent in rebuttal and two reviewers accordingly revised their scores. 3 and 4 are valid points but do not seem to hinder the significance of the suggested approach (e.g., scale can happen in follow up paper). Overall this paper seems to be in the weak accept regime.